# A2BCD: Asynchronous Acceleration with Optimal Complexity

**Robert Hannah,**[*] **Fei Feng,**[†] **Wotao Yin**[‡]
Department of Mathematics
University of California, Los Angeles
520 Portola Plaza, Los Angeles, CA 90095, USA

## Abstract

In this paper, we propose the **A**synchronous **A**ccelerated Nonuniform Randomized **B**lock **C**oordinate **D**escent algorithm (`A2BCD`). We prove `A2BCD` converges linearly to a solution of the convex minimization problem at the same rate as `NU_ACDM`, so long as the maximum delay is not too large. This is the first asynchronous Nesterov-accelerated algorithm that attains any provable speedup. Moreover, we then prove that these algorithms both have optimal complexity. Asynchronous algorithms complete much faster iterations, and `A2BCD` has optimal complexity. Hence we observe in experiments that `A2BCD` is the top-performing coordinate descent algorithm, converging up to $4 - 5\times$ faster than `NU_ACDM` on some data sets in terms of wall-clock time. To motivate our theory and proof techniques, we also derive and analyze a continuous-time analogue of our algorithm and prove it converges at the same rate.

## 1 Introduction

In this paper, we propose and prove the convergence of the **A**synchronous **A**ccelerated Nonuniform Randomized **B**lock **C**oordinate **D**escent algorithm (`A2BCD`), the first asynchronous Nesterov-accelerated algorithm that achieves optimal complexity. No previous attempts have been able to prove a speedup for asynchronous Nesterov acceleration. We aim to find the minimizer $x_*$ of the unconstrained minimization problem:

$$\min_{x \in \mathbb{R}^d} f(x) = f\big(x_{(1)}, \ldots, x_{(n)}\big) \tag{1.1}$$

where $f$ is $\sigma$-strongly convex for $\sigma > 0$ with $L$-Lipschitz gradient $\nabla f = (\nabla_1 f, \ldots, \nabla_n f)$. $x \in \mathbb{R}^d$ is composed of coordinate blocks $x_{(1)}, \ldots, x_{(n)}$. The coordinate blocks of the gradient $\nabla_i f$ are assumed $L_i$-Lipschitz with respect to the $i$th block. That is, $\forall x, h \in \mathbb{R}^d$:

$$\|\nabla_i f(x + P_i h) - \nabla_i f(x)\| \leq L_i \|h\| \tag{1.2}$$

where $P_i$ is the projection onto the $i$th block of $\mathbb{R}^d$. Let $\bar{L} \triangleq \frac{1}{n} \sum_{i=1}^{n} L_i$ be the average block Lipschitz constant. These conditions on $f$ are assumed throughout this whole paper. Our algorithm can also be applied to non-strongly convex objectives ($\sigma = 0$) or non-smooth objectives using the *black box reduction* techniques proposed in Allen-Zhu & Hazan (2016). Hence we consider only

---

[*]Corresponding author: RobertHannah89@gmail.com
[†]fei.feng@math.ucla.edu
[‡]wotaoyin@math.ucla.edu

the coordinate smooth, strongly-convex case. Our algorithm can also be applied to the convex regularized ERM problem via the standard dual transformation (see for instance Lin et al. (2014)):

$$f(x) = \frac{1}{n} \sum_{i=1}^{n} f_i(\langle a_i, x \rangle) + \frac{\lambda}{2} \|x\|^2 \tag{1.3}$$

Hence `A2BCD` can be used as an asynchronous Nesterov-accelerated finite-sum algorithm.

Coordinate descent methods, in which a chosen coordinate block $i_k$ is updated at every iteration, are a popular way to solve equation 1.1. Randomized block coordinate descent (`RBCD`, Nesterov (2012)) updates a uniformly randomly chosen coordinate block $i_k$ with a gradient-descent-like step: $x_{k+1} = x_k - (1/L_{i_k}) \nabla_{i_k} f(x_k)$. The complexity $K(\epsilon)$ of an algorithm is defined as the number of iterations required to decrease the error $\mathbb{E}(f(x_k) - f(x_*))$ to less than $\epsilon(f(x_0) - f(x_*))$. Randomized coordinate descent has a complexity of $K(\epsilon) = \mathcal{O}(n(\bar{L}/\sigma) \ln(1/\epsilon))$.

Using a series of averaging and extrapolation steps, *accelerated* `RBCD` Nesterov (2012) improves `RBCD`'s iteration complexity $K(\epsilon)$ to $\mathcal{O}(n\sqrt{\bar{L}/\sigma} \ln(1/\epsilon))$, which leads to much faster convergence when $\frac{\bar{L}}{\sigma}$ is large. This rate is optimal when all $L_i$ are equal Lan & Zhou (2015). Finally, using a special probability distribution for the random block index $i_k$, the non-uniform accelerated coordinate descent method Allen-Zhu et al. (2015) (`NU_ACDM`) can further decrease the complexity to $\mathcal{O}(\sum_{i=1}^{n} \sqrt{L_i/\sigma} \ln(1/\epsilon))$, which can be up to $\sqrt{n}$ times faster than accelerated `RBCD`, since some $L_i$ can be significantly smaller than $L$. `NU_ACDM` is the current state-of-the-art coordinate descent algorithm for solving equation 1.1.

Our `A2BCD` algorithm generalizes `NU_ACDM` to the asynchronous-parallel case. We solve equation 1.1 with a collection of $p$ computing nodes that continually read a shared-access solution vector $y$ into local memory then compute a block gradient $\nabla_i f$, which is used to update shared solution vectors $(x, y, v)$. Proving convergence in the asynchronous case requires extensive new technical machinery.

A traditional synchronous-parallel implementation is organized into rounds of computation: Every computing node must complete an update in order for the next iteration to begin. However, this synchronization process can be extremely costly, since the lateness of a single node can halt the entire system. This becomes increasingly problematic with scale, as differences in node computing speeds, load balancing, random network delays, and bandwidth constraints mean that a synchronous-parallel solver may spend more time waiting than computing a solution.

Computing nodes in an asynchronous solver do not wait for others to complete and share their updates before starting the next iteration. They simply continue to update the solution vectors with the most recent information available, without any central coordination. This eliminates costly idle time, meaning that asynchronous algorithms can be much faster than traditional ones, since they have much faster iterations. For instance, random network delays cause asynchronous algorithms to complete iterations $\Omega(\ln(p))$ time faster than synchronous algorithms at scale. This and other factors that influence the speed of iterations are discussed in Hannah & Yin (2017a). However, since many iterations may occur between the time that a node reads the solution vector, and the time that its computed update is applied, effectively the solution vector is being updated with outdated information. At iteration $k$, the block gradient $\nabla_{i_k} f$ is computed at a *delayed iterate* $\hat{y}_k$ defined as[1]:

$$\hat{y}_k = \left( \left( y_{k-j(k,1)} \right)_{(1)}, \ldots, \left( y_{k-j(k,n)} \right)_{(n)} \right) \tag{1.4}$$

---

[1]Every coordinate can be outdated by a different amount without significantly changing the proofs.

for delay parameters $j(k, 1), \ldots, j(k, n) \in \mathbb{N}$. Here $j(k, i)$ denotes how many iterations out of date coordinate block $i$ is at iteration $k$. Different blocks may be out of date by different amounts, which is known as an *inconsistent read*. We assume[2] that $j(k, i) \leq \tau$ for some constant $\tau < \infty$.

Asynchronous algorithms were proposed in Chazan & Miranker (1969) to solve linear systems. General convergence results and theory were developed later in Bertsekas (1983); Bertsekas & Tsitsiklis (1997); Tseng et al. (1990); Luo & Tseng (1992; 1993); Tseng (1991) for partially and totally asynchronous systems, with essentially-cyclic block sequence $i_k$. More recently, there has been renewed interest in asynchronous algorithms with random block coordinate updates. Linear and sublinear convergence results were proven for asynchronous RBCD Liu & Wright (2015); Liu et al. (2014); Avron et al. (2014), and similar was proven for asynchronous SGD Recht et al. (2011), and variance reduction algorithms Reddi et al. (2015); Leblond et al. (2017); Mania et al. (2015); Huo & Huang (2016), and primal-dual algorithms Combettes & Eckstein (2018).

There is also a rich body of work on asynchronous SGD. In the distributed setting, Zhou et al. (2018) showed global convergence for stochastic variationally coherent problems even when the delays grow at a polynomial rate. In Lian et al. (2018), an asynchronous decentralized SGD was proposed with the same optimal sublinear convergence rate as SGD and linear speedup with respect to the number of workers. In Liu et al. (2018), authors obtained an asymptotic rate of convergence for asynchronous momentum SGD on streaming PCA, which provides insight into the tradeoff between asynchrony and momentum. In Dutta et al. (2018), authors prove convergence results for asynchronous SGD that highlight the tradeoff between faster iterations and iteration complexity. Further related work is discussed in Section 4.

## 1.1  Summary of Contributions

In this paper, we prove that `A2BCD` attains `NU_ACDM`'s state-of-the-art iteration complexity to highest order for solving equation 1.1, so long as delays are not too large (see Section 2). The proof is very different from that of Allen-Zhu et al. (2015), and involves significant technical innovations and complexity related to the analysis of asynchronicity.

We also prove that `A2BCD` (and hence `NU_ACDM`) has optimal complexity to within a constant factor over a fairly general class of randomized block coordinate descent algorithms (see Section 2.1). This extends results in Lan & Zhou (2015) to asynchronous algorithms with $L_i$ not all equal. Since asynchronous algorithms complete faster iterations, and `A2BCD` has optimal complexity, we expect `A2BCD` to be faster than all existing coordinate descent algorithms. We confirm with numerical experiments that `A2BCD` is the current fastest coordinate descent algorithm (see Section 5).

We are only aware of one previous and one contemporaneous attempt at proving convergence results for asynchronous Nesterov-accelerated algorithms. However, the first is not accelerated and relies on extreme assumptions, and the second obtains no speedup. Therefore, we claim that our results are the first-ever analysis of asynchronous Nesterov-accelerated algorithms that attains a speedup. Moreover, our speedup is optimal for delays not too large[3].

The work of Meng et al. claims to obtain square-root speedup for an asynchronous accelerated SVRG. In the case where all component functions have the same Lipschitz constant $L$, the complexity they obtain reduces to $(n + \kappa) \ln(1/\epsilon)$ for $\kappa = \mathcal{O}(\tau n^2)$ (Corollary 4.4). Hence authors do not even obtain accelerated rates. Their convergence condition is $\tau < \frac{1}{4\Delta^{1/8}}$ for sparsity parameter $\Delta$. Since the dimension $d$ satisfies $d \geq \frac{1}{\Delta}$, they require $d \geq 2^{16}\tau^8$. So $\tau = 20$ requires dimension $d > 10^{15}$.

---

[2]This condition can be relaxed however by techniques in Hannah & Yin (2017b); Sun et al. (2017); Peng et al. (2016c); Hannah & Yin (2017a)

[3]Speedup is defined precisely in Section 2

In a contemporaneous preprint, authors in Fang et al. (2018) skillfully devised accelerated schemes for asynchronous coordinate descent and SVRG using momentum compensation techniques. Although their complexity results have the improved $\sqrt{\kappa}$ dependence on the condition number, they do not prove any speedup. Their complexity is $\tau$ times larger than the serial complexity. Since $\tau$ is necessarily greater than $p$, their results imply that adding more computing nodes will increase running time. The authors claim that they can extend their results to linear speedup for asynchronous, accelerated SVRG under sparsity assumptions. And while we think this is quite likely, they have not yet provided proof.

We also derive a second-order ordinary differential equation (ODE), which is the continuous-time limit of A2BCD (see Section 3). This extends the ODE found in Su et al. (2014) to an *asynchronous* accelerated algorithm minimizing a *strongly convex* function. We prove this ODE linearly converges to a solution with the same rate as A2BCD's, without needing to resort to the restarting techniques. The ODE analysis motivates and clarifies the our proof strategy of the main result.

## 2 MAIN RESULTS

We should consider functions $f$ where it is efficient to calculate blocks of the gradient, so that coordinate-wise parallelization is efficient. That is, the function should be "coordinate friendly" Peng et al. (2016b). This is a very wide class that includes regularized linear regression, logistic regression, etc. The $L^2$-regularized empirical risk minimization problem is not coordinate friendly in general, however the equivalent dual problem is, and hence can be solved efficiently by A2BCD (see Lin et al. (2014), and Section 5).

To calculate the $k + 1$'th iteration of the algorithm from iteration $k$, we use only one block of the gradient $\nabla_{i_k} f$. We assume that the delays $j(k, i)$ are independent of the block sequence $i_k$, but otherwise arbitrary (This is a standard assumption found in the vast majority of papers, but can be relaxed Sun et al. (2017); Leblond et al. (2017); Cannelli et al. (2017)).

**Definition 1. Asynchronous Accelerated Randomized Block Coordinate Descent (A2BCD).** Let $f$ be $\sigma$-strongly convex, and let its gradient $\nabla f$ be $L$-Lipschitz with block coordinate Lipschitz parameters $L_i$ as in equation 1.2. We define the **condition number** $\kappa = L/\sigma$, and let $\underline{L} = \min_i L_i$. Using these parameters, we sample $i_k$ in an independent and identically distributed (IID) fashion according to

$$\mathbb{P}[i_k = j] = L_j^{1/2}/S, \quad j \in \{1, \ldots, n\}, \quad \text{for } S = \sum_{i=1}^{n} L_i^{1/2}. \tag{2.1}$$

Let $\tau$ be the maximum asynchronous delay. We define the dimensionless **asynchronicity parameter** $\psi$, which is proportional to $\tau$, and quantifies how strongly asynchronicity will affect convergence:

$$\psi = 9\left(S^{-1/2}\underline{L}^{-1/2}L^{3/4}\kappa^{1/4}\right) \times \tau \tag{2.2}$$

We use the above system parameters and $\psi$ to define the coefficients $\alpha, \beta$, and $\gamma$ via eqs. (2.3) to (2.5). Hence A2BCD algorithm is defined via the iterations: eqs. (2.6) to (2.8).

$$\alpha \triangleq \left(1 + (1 + \psi)\sigma^{-1/2}S\right)^{-1} \tag{2.3}$$

$$\beta \triangleq 1 - (1 - \psi)\sigma^{1/2}S^{-1} \tag{2.4}$$

$$h \triangleq 1 - \frac{1}{2}\sigma^{1/2}\underline{L}^{-1/2}\psi. \tag{2.5}$$

$$y_k = \alpha v_k + (1 - \alpha)x_k, \tag{2.6}$$

$$x_{k+1} = y_k - hL_{i_k}^{-1}\nabla_{i_k} f(\hat{y}_k), \tag{2.7}$$

$$v_{k+1} = \beta v_k + (1 - \beta)y_k - \sigma^{-1/2}L_{i_k}^{-1/2}\nabla_{i_k} f(\hat{y}_k). \tag{2.8}$$

See Section A for a discussion of why it is practical and natural to have the gradient $\nabla_{i_k} f(\hat{y}_k)$ to be outdated, while the actual variables $x_k, y_k, v_k$ can be efficiently kept up to date. Essentially it is

because most of the computation lies in computing $\nabla_{i_k} f(\hat{y}_k)$. After this is computed, $x_k, y_k, v_k$ can be updated more-or-less atomically with minimal overhead, meaning that they will always be up to date. However our main results still hold for more general asynchronicity.

A natural quantity to consider in asynchronous convergence analysis is the **asynchronicity error**, a powerful tool for analyzing asynchronous algorithms used in several recent works Peng et al. (2016a); Hannah & Yin (2017b); Sun et al. (2017); Hannah & Yin (2017a). We adapt it and use a weighted sum of the history of the algorithm with decreasing weight as you go further back in time.

**Definition 2. Asynchronicity error.** Using the above parameters, we define:

$$A_k = \sum_{j=1}^{\tau} c_j \|y_{k+1-j} - y_{k-j}\|^2 \quad (2.9) \quad \text{for } c_i = \frac{6}{S} L^{1/2} \kappa^{3/2} \tau \sum_{j=i}^{\tau} \left(1 - \sigma^{1/2} S^{-1}\right)^{i-j-1} \psi^{-1}. \quad (2.10)$$

Here we define $y_k = y_0$ for all $k < 0$. The determination of the coefficients $c_i$ is in general a very involved process of trial and error, intuition, and balancing competing requirements. The algorithm doesn't depend on the coefficients, however; they are only an analytical tool.

We define $\mathbb{E}_k[X]$ as the expectation of $X$ conditioned on $(x_0, \ldots, x_k)$, $(y_0, \ldots, y_k)$, $(v_0, \ldots, v_k)$, and $(i_0, \ldots, i_{k-1})$. To simplify notation[4], we assume that the minimizer $x_* = 0$, and that $f(x_*) = 0$ with no loss in generality. We define the **Lyapunov function**:

$$\rho_k = \|v_k\|^2 + A_k + cf(x_k) \qquad (2.11) \qquad\qquad \text{for } c = 2\sigma^{-1/2} S^{-1} \left(\beta \alpha^{-1}(1 - \alpha) + 1\right). \qquad (2.12)$$

We now present this paper's first main contribution.

**Theorem 1.** Let $f$ be $\sigma$-strongly convex with a gradient $\nabla f$ that is $L$-Lipschitz with block Lipschitz constants $\{L_i\}_{i=1}^{n}$. Let $\psi$ defined in equation 2.2 satisfy $\psi \leq \frac{3}{7}$ (i.e. $\tau \leq \frac{1}{21} S^{1/2} \underline{L}^{1/2} L^{-3/4} \kappa^{-1/4}$). Then for A2BCD we have:

$$\mathbb{E}_k[\rho_{k+1}] \leq \left(1 - (1 - \psi)\sigma^{1/2} S^{-1}\right)\rho_k.$$

To obtain $\mathbb{E}[\rho_k] \leq \epsilon \rho_0$, it takes $K_{\text{A2BCD}}(\epsilon)$ iterations for:

$$K_{\text{A2BCD}}(\epsilon) = \left(\sigma^{-1/2} S + \mathcal{O}(1)\right)\frac{\ln(1/\epsilon)}{1 - \psi}, \qquad (2.13)$$

where $\mathcal{O}(\cdot)$ is asymptotic with respect to $\sigma^{-1/2} S \to \infty$, and uniformly bounded.

This result is proven in Section B. A stronger result for $L_i \equiv L$ can be proven, but this adds to the complexity of the proof; see Section E for a discussion. In practice, asynchronous algorithms are far more resilient to delays than the theory predicts. $\tau$ can be much larger without negatively affecting the convergence rate and complexity. This is perhaps because we are limited to a worst-case analysis, which is not representative of the average-case performance.

Allen-Zhu et al. (2015) (Theorem 5.1) shows a linear convergence rate of $1 - 2/\left(1 + 2\sigma^{-1/2} S\right)$ for NU_ACDM, which leads to the corresponding iteration complexity of $K_{\text{NU\_ACDM}}(\epsilon) = \left(\sigma^{-1/2} S + \mathcal{O}(1)\right) \ln(1/\epsilon)$. Hence, we have:

$$K_{\text{A2BCD}}(\epsilon) = \frac{1}{1 - \psi}(1 + o(1)) K_{\text{NU\_ACDM}}(\epsilon)$$

---

[4]We can assume $x_* = 0$ with no loss in generality since we may translate the coordinate system so that $x_*$ is at the origin. We can assume $f(x_*) = 0$ with no loss in generality, since we can replace $f(x)$ with $f(x) - f(x_*)$. Without this assumption, the Lyapunov function simply becomes: $\|v_k - x_*\|^2 + A_k + c(f(x_k) - f(x_*))$.

When $0 \leq \psi \ll 1$, or equivalently, when $\tau \ll S^{1/2} \underline{\mathbf{L}}^{1/2} L^{-3/4} \kappa^{-1/4}$, the complexity of A2BCD asymptotically matches that of NU_ACDM. Hence A2BCD combines state-of-the-art complexity with the faster iterations and superior scaling that asynchronous iterations allow. We now present some special cases of the conditions on the maximum delay $\tau$ required for good complexity.

**Corollary 3.** *Let the conditions of Theorem 1 hold. If all coordinate-wise Lipschitz constants $L_i$ are equal (i.e. $L_i = L_1$, $\forall i$), then we have $K_{\text{A2BCD}}(\epsilon) \sim K_{\text{NU\_ACDM}}(\epsilon)$ when $\tau \ll n^{1/2}\kappa^{-1/4}(L_1/L)^{3/4}$. If we further assume all coordinate-wise Lipschitz constants $L_i$ equal $L$. Then $K_{\text{A2BCD}}(\epsilon) \sim K_{\text{NU\_ACDM}}(\epsilon) = K_{\text{ACDM}}(\epsilon)$, when $\tau \ll n^{1/2}\kappa^{-1/4}$.*

**Remark 1. Reduction to synchronous case.** Notice that when $\tau = 0$, we have $\psi = 0$, $c_i \equiv 0$ and hence $A_k \equiv 0$. Thus A2BCD becomes equivalent to NU_ACDM, the Lyapunov function[5] $\rho_k$ becomes equivalent to one found in Allen-Zhu et al. (2015)(pg. 9), and Theorem 1 yields the same complexity.

The maximum delay $\tau$ will be a function $\tau(p)$ of $p$, number of computing nodes. Clearly $\tau \geq p$, and experimentally it has been observed that $\tau = \mathcal{O}(p)$ Leblond et al. (2017). Let gradient complexity $K(\epsilon, \tau)$ be the number of gradients required for an asynchronous algorithm with maximum delay $\tau$ to attain suboptimality $\epsilon$. $\tau(1) = 0$, since with only 1 computing node there can be no delay. This corresponds to the serial complexity. We say that an asynchronous algorithm attains a *complexity speedup* if $\frac{pK(\epsilon,\tau(0))}{K(\epsilon,\tau(p))}$ is increasing in $p$. We say it attains *linear complexity speedup* if $\frac{pK(\epsilon,\tau(0))}{K(\epsilon,\tau(p))} = \Omega(p)$. In Theorem 1, we obtain a linear complexity speedup (for $p$ not too large), whereas no other prior attempt can attain even a complexity speedup with Nesterov acceleration.

In the ideal scenario where the rate at which gradients are calculated increases linearly with $p$, algorithms that have linear complexity speedup will have a linear decrease in wall-clock time. However in practice, when the number of computing nodes is sufficiently large, the rate at which gradients are calculated will no longer be linear. This is due to many parallel overhead factors including too many nodes sharing the same memory read/write bandwidth, and network bandwidth. However we note that even with these issues, we obtain much faster convergence than the synchronous counterpart experimentally.

## 2.1  Optimality

NU_ACDM and hence A2BCD are in fact optimal in some sense. That is, among a fairly wide class of coordinate descent algorithms $\mathcal{A}$, they have the best-possible worst-case complexity to highest order. We extend the work in Lan & Zhou (2015) to encompass algorithms are asynchronous and have unequal $L_i$. For a subset $S \in \mathbb{R}^d$, we let IC($S$) (inconsistent read) denote the set of vectors $v$ whose components are a combination of components of vectors in the set $S$. That is, $v = (v_{1,1}, v_{2,2}, \ldots, v_{d,d})$ for some vectors $v_1, v_2, \ldots, v_d \in S$. Here $v_{i,j}$ denotes the $j$th component of vector $v_i$.

**Definition 4. Asynchronous Randomized Incremental Algorithms.** Consider the unconstrained minimization problem equation 1.1 for function $f$ satisfying the conditions stated in Section 1. We define the class $\mathcal{A}$ as algorithms $G$ on this problem such that:

1. For each parameter set $(\sigma, L_1, \ldots, L_n, n)$, $G$ has an associated IID random variable $i_k$ with some fixed distribution $\mathbb{P}[i_k] = p_i$ for $\sum_{i=1}^n p_i = 1$.

2. The iterates of $A$ satisfy: $x_{k+1} \in \text{span}\{\text{IC}(X_k), \nabla_{i_0} f(\text{IC}(X_0)), \nabla_{i_1} f(\text{IC}(X_1)), \ldots, \nabla_{i_k} f(\text{IC}(X_k))\}$

This is a rather general class: $x_{k+1}$ can be constructed from any inconsistent reading of past iterates IC($X_k$), and any past gradient of an inconsistent read $\nabla_{i_j} f(\text{IC}(X_j))$.

---

[5]Their Lyapunov function is in fact a generalization of the one found in Nesterov (2012).

**Theorem 2.** *For any algorithm $G \in \mathcal{A}$ that solves eq. (1.1), and parameter set $(\sigma, L_1, \ldots, L_n, n)$, there is a dimension $d$, a corresponding function $f$ on $\mathbb{R}^d$, and a starting point $x_0$, such that*

$$\mathbb{E}\|x_k - x_*\|^2 / \|x_0 - x_*\|^2 \geq \frac{1}{2}\big(1 - 4/\big(\sum_{j=1}^n \sqrt{L_i/\sigma} + 2n\big)\big)^k$$

*Hence $\mathcal{A}$ has a complexity lower bound: $K(\epsilon) \geq \frac{1}{4}(1 + o(1))\big(\sum_{j=1}^n \sqrt{L_i/\sigma} + 2n\big)\ln(1/2\epsilon)$*

Our proof in Section D follows very similar lines to Lan & Zhou (2015); Nesterov (2013).

## 3 ODE ANALYSIS

In this section we present and analyze an ODE which is the continuous-time limit of `A2BCD`. This ODE is a strongly convex, and asynchronous version of the ODE found in Su et al. (2014). For simplicity, assume $L_i = L$, $\forall i$. We rescale (I.e. we replace $f(x)$ with $\frac{1}{\sigma}f$.) $f$ so that $\sigma = 1$, and hence $\kappa = L/\sigma = L$. Taking the discrete limit of synchronous `A2BCD` (i.e. accelerated `RBCD`), we can derive the following ODE[6] (see Section equation C.1):

$$\ddot{Y} + 2n^{-1}\kappa^{-1/2}\dot{Y} + 2n^{-2}\kappa^{-1}\nabla f(Y) = 0 \tag{3.1}$$

We define the parameter $\eta \triangleq n\kappa^{1/2}$, and the energy: $E(t) = e^{n^{-1}\kappa^{-1/2}t}(f(Y) + \frac{1}{4}\|Y + \eta\dot{Y}\|^2)$. This is very similar to the Lyapunov function discussed in equation 2.11, with $\frac{1}{4}\|Y(t) + \eta\dot{Y}(t)\|^2$ fulfilling the role of $\|v_k\|^2$, and $A_k = 0$ (since there is no delay yet). Much like the traditional analysis in the proof of Theorem 1, we can derive a linear convergence result with a similar rate. See Section C.2.

**Lemma 5.** *If $Y$ satisfies equation 3.1, the energy satisfies $E'(t) \leq 0$, $E(t) \leq E(0)$, and hence:*

$$f(Y(t)) + \frac{1}{4}\left\|Y(t) + n\kappa^{1/2}\dot{Y}(t)\right\|^2 \leq \left(f(Y(0)) + \frac{1}{4}\left\|Y(0) + \eta\dot{Y}(0)\right\|^2\right)e^{-n^{-1}\kappa^{-1/2}t}$$

We may also analyze an asynchronous version of equation 3.1 to motivate the proof of our main theorem. Here $\hat{Y}(t)$ is a delayed version of $Y(t)$ with the delay bounded by $\tau$.

$$\ddot{Y} + 2n^{-1}\kappa^{-1/2}\dot{Y} + 2n^{-2}\kappa^{-1}\nabla f\big(\hat{Y}\big) = 0, \tag{3.2}$$

Unfortunately, this energy satisfies (see Section equation C.4, equation C.7):

$$e^{-\eta^{-1}t}E'(t) \leq -\frac{1}{8}\eta\left\|\dot{Y}\right\|^2 + 3\kappa^2\eta^{-1}\tau D(t), \text{ for } D(t) \triangleq \int_{t-\tau}^t \left\|\dot{Y}(s)\right\|^2 ds.$$

Hence this energy $E(t)$ may not be decreasing in general. But, we may add a continuous-time **asynchronicity error** (see Sun et al. (2017)), much like in Definition 2, to create a decreasing energy. Let $c_0 \geq 0$ and $r > 0$ be arbitrary constants that will be set later. Define:

$$A(t) = \int_{t-\tau}^t c(t-s)\left\|\dot{Y}(s)\right\|^2 ds, \text{ for } c(t) \triangleq c_0\left(e^{-rt} + \frac{e^{-r\tau}}{1 - e^{-r\tau}}\big(e^{-rt} - 1\big)\right).$$

**Lemma 6.** *When $r\tau \leq \frac{1}{2}$, the asynchronicity error $A(t)$ satisfies:*

$$e^{-rt}\frac{d}{dt}\big(e^{rt}A(t)\big) \leq c_0\left\|\dot{Y}(t)\right\|^2 - \frac{1}{2}\tau^{-1}c_0 D(t).$$

---

[6]For compactness, we have omitted the $(t)$ from time-varying functions $Y(t)$, $\dot{Y}(t)$, $\nabla Y(t)$, etc.

See Section C.3 for the proof. Adding this error to the Lyapunov function serves a similar purpose in the continuous-time case as in the proof of Theorem 1 (see Lemma 11). It allows us to negate $\frac{1}{2}\tau^{-1}c_0$ units of $D(t)$ for the cost of creating $c_0$ units of $\left\|\dot{Y}(t)\right\|^2$. This restores monotonicity.

**Theorem 3.** *Let $c_0 = 6\kappa^2\eta^{-1}\tau^2$, and $r = \eta^{-1}$. If $\tau \leq \frac{1}{\sqrt{48}}n\kappa^{-1/2}$ then we have:*

$$e^{-\eta^{-1}t}\frac{d}{dt}\Big(E(t) + e^{\eta^{-1}t}A(t)\Big) \leq 0. \tag{3.3}$$

*Hence $f(Y(t))$ convergence linearly to $f(x_*)$ with rate $\mathcal{O}\big(\exp\big(-t/(n\kappa^{1/2})\big)\big)$*

Notice how this convergence condition is similar to Corollary 3, but a little looser. The convergence condition in Theorem 1 can actually be improved to approximately match this (see Section E).

*Proof.*
$$e^{-\eta^{-1}t}\frac{d}{dt}\Big(E(t) + e^{\eta^{-1}t}A(t)\Big) \leq \left(c_0 - \frac{1}{8}\eta\right)\left\|\dot{Y}\right\|^2 + \left(3\kappa^2\eta^{-1}\tau - \frac{1}{2}\tau^{-1}c_0\right)D(t)$$
$$= 6\eta^{-1}\kappa^2\left(\tau^2 - \frac{1}{48}n^2\kappa^{-1}\right)\left\|\dot{Y}\right\|^2 \leq 0 \qquad \square$$

The preceding should hopefully elucidate the logic and general strategy of the proof of Theorem 1.

## 4 RELATED WORK

We now discuss related work that was not addressed in Section 1. Nesterov acceleration is a method for improving an algorithm's iteration complexity's dependence the condition number $\kappa$. Nesterov-accelerated methods have been proposed and discovered in many settings Nesterov (1983); Tseng (2008); Nesterov (2012); Lin et al. (2014); Lu & Xiao (2014); Shalev-Shwartz & Zhang (2016); Allen-Zhu (2017), including for coordinate descent algorithms (algorithms that use 1 gradient block $\nabla_i f$ or minimize with respect to 1 coordinate block per iteration), and incremental algorithms (algorithms for finite sum problems $\frac{1}{n}\sum_{i=1}^{n} f_i(x)$ that use 1 function gradient $\nabla f_i(x)$ per iteration). Such algorithms can often be augmented to solve composite minimization problems (minimization for objective of the form $f(x) + g(x)$, especially for nonsmooth $g$), or include constraints.

In Peng et al. (2016a), authors proposed and analyzed an asynchronous fixed-point algorithm called ARock, that takes proximal algorithms, forward-backward, ADMM, etc. as special cases. Work has also been done on asynchronous algorithms for finite sums in the operator setting Davis (2016); Johnstone & Eckstein (2018). In Hannah & Yin (2017b); Sun et al. (2017); Peng et al. (2016c); Cannelli et al. (2017) showed that many of the assumptions used in prior work (such as bounded delay $\tau < \infty$) were unrealistic and unnecessary in general. In Hannah & Yin (2017a) the authors showed that asynchronous iterations will complete far more iterations per second, and that a wide class of asynchronous algorithms, including asynchronous `RBCD`, have the same iteration complexity as their synchronous counterparts. Hence certain asynchronous algorithms can be expected to significantly outperform traditional ones.

In Xiao et al. (2017) authors propose a novel asynchronous catalyst-accelerated Lin et al. (2015) primal-dual algorithmic framework to solve regularized ERM problems. They structure the parallel updates so that the data that an update depends on is up to date (though the rest of the data may not be). However catalyst acceleration incurs a $\log(\kappa)$ penalty over Nesterov acceleration in general. In Allen-Zhu (2017), the author argues that the inner iterations of catalyst acceleration are hard to tune, making it less practical than Nesterov acceleration.

## 5 NUMERICAL EXPERIMENTS

To investigate the performance of `A2BCD`, we solve the ridge regression problem. Consider the following primal and corresponding dual objective (see for instance Lin et al. (2014)):

$$\min_{w \in \mathbb{R}^d} P(w) = \frac{1}{2n}\left\|A^T w - l\right\|^2 + \frac{\lambda}{2}\|w\|^2, \min_{\alpha \in \mathbb{R}^n} D(\alpha) = \frac{1}{2d^2\lambda}\|A\alpha\|^2 + \frac{1}{2d}\|\alpha + l\|^2 \qquad (5.1)$$

where $A \in \mathbb{R}^{d \times n}$ is a matrix of $n$ samples and $d$ features, and $l$ is a label vector. We let $A = [A_1, \ldots, A_m]$ where $A_i$ are the column blocks of $A$. We compare `A2BCD` (which is asynchronous accelerated), synchronous `NU_ACDM` (which is synchronous accelerated), and asynchronous `RBCD` (which is asynchronous non-accelerated). Nodes randomly select a coordinate block according to equation 2.1, calculate the corresponding block gradient, and use it to apply an update to the shared solution vectors. synchronous `NU_ACDM` is implemented in a batch fashion, with batch size $p$ (1 block per computing node). Nodes in synchronous `NU_ACDM` implementation must wait until all nodes apply their computed gradients before they can start the next iteration, but the asynchronous algorithms simply compute with the most up-to-date information available.

We use the datasets `w1a` (47272 samples, 300 features), `wxa` which combines the data from from `w1a` to `w8a` (293201 samples, 300 features), and `aloi` (108000 samples, 128 features) from LIBSVM Chang & Lin (2011). The algorithm is implemented in a multi-threaded fashion using C++11 and GNU Scientific Library with a shared memory architecture. We use 40 threads on two 2.5GHz 10-core Intel Xeon E5-2670v2 processors. See Section A.1 for a discussion of parameter tuning and estimation. The parameters for each algorithm are tuned to give the fastest performance, so that a fair comparison is possible.

A critical ingredient in the efficient implementation of `A2BCD` and `NU_ACDM` for this problem is the efficient update scheme discussed in Lee & Sidford (2013b;a). In linear regression applications such as this, it is essential to be able to efficiently maintain or recover $Ay$. This is because calculating block gradients requires the vector $A_i^T Ay$, and without an efficient way to recover $Ay$, block gradient evaluations are essentially 50% as expensive as full-gradient calculations. Unfortunately, every accelerated iteration results in dense updates to $y_k$ because of the averaging step in equation 2.6. Hence $Ay$ must be recalculated from scratch.

However Lee & Sidford (2013a) introduces a linear transformation that allows for an equivalent iteration that results in sparse updates to new iteration variables $p$ and $q$. The original purpose of this transformation was to ensure that the averaging steps (e.g. equation 2.6) do not dominate the computational cost for sparse problems. However we find a more important secondary use which applies to both sparse and dense problems. Since the updates to $p$ and $q$ are sparse coordinate-block updates, the vectors $Ap$, and $Aq$ can be efficiently maintained, and therefore block gradients can be efficiently calculated. The specifics of this efficient implementation are discussed in Section A.2.

In Table 5, we plot the sub-optimality vs. time for decreasing values of $\lambda$, which corresponds to increasingly large condition numbers $\kappa$. When $\kappa$ is small, acceleration doesn't result in a significantly better convergence rate, and hence `A2BCD` and async-RBCD both outperform sync-`NU_ACDM` since they complete faster iterations at similar complexity. Acceleration for low $\kappa$ has unnecessary overhead, which means async-RBCD can be quite competitive. When $\kappa$ becomes large, async-RBCD is no longer competitive, since it has a poor convergence rate. We observe that `A2BCD` and sync-`NU_ACDM` have essentially the same convergence rate, but `A2BCD` is up to $4 - 5\times$ faster than sync-`NU_ACDM` because it completes much faster iterations. We observe this advantage despite the fact that we are in an ideal environment for synchronous computation: A small, homogeneous, high-bandwidth, low-latency cluster. In large-scale heterogeneous systems with greater synchronization overhead, bandwidth constraints, and latency, we expect `A2BCD`'s advantage to be much larger.

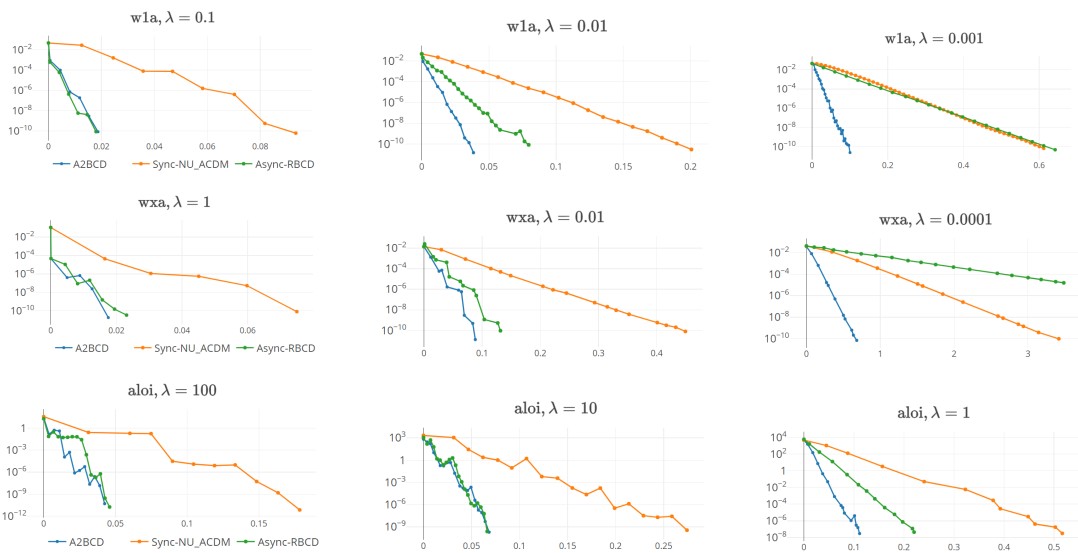

Table 1: Sub-optimality $f(y_k) - f(x_*)$ (y-axis) vs time in seconds (x-axis) for `A2BCD`, synchronous `NU_ACDM`, and asynchronous RBCD for data sets `w1a`, `wxa` and `aloi` for various values of $\lambda$.

## 6 ACKNOWLEDGEMENT

The authors would like to thank the reviewers for their helpful comments. The research presented in this paper was supported in part by AFOSR MURI FA9550-18-10502, NSF DMS-1720237, and ONR N0001417121.

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

## A    Efficient Implementation

An efficient implementation will have coordinate blocks of size greater than 1. This to ensure the efficiency of linear algebra subroutines. Especially because of this, the bulk of the computation for each iteration is computing $\nabla_{i_k} f(\hat{y}_k)$, and not the averaging steps. Hence the computing nodes only need a local copy of $y_k$ in order to do the bulk of an iteration's computation. Given this gradient $\nabla_{i_k} f(\hat{y}_k)$, updating $y_k$ and $v_k$ is extremely fast ($x_k$ can simply be eliminated). Hence it is natural to simply store $y_k$ and $v_k$ centrally, and update them when the delayed gradients $\nabla_{i_k} f(\hat{y}_k)$. Given the above, a write mutex over $(y, v)$ has minuscule overhead (which we confirm with experiments), and makes the labeling of iterates unambiguous. This also ensures that $v_k$ and $y_k$ *are always up to date* when $(y, v)$ are being updated. Whereas the gradient $\nabla_{i_k} f(\hat{y}_k)$ may at the same time be out of date, since it has been calculated with an outdated version of $y_k$. However a write mutex is not necessary in practice, and does not appear to affect convergence rates or computation time. Also it is possible to prove convergence under more general asynchronicity.

### A.1    Parameter selection and tuning

When defining the coefficients, $\sigma$ may be underestimated, and $L, L_1, \ldots, L_n$ may be overestimated if exact values are unavailable. Notice that $x_k$ can be eliminated from the above iteration, and the block gradient $\nabla_{i_k} f(\hat{y}_k)$ only needs to be calculated once per iteration. A larger (or overestimated) maximum delay $\tau$ will cause a larger asynchronicity parameter $\psi$, which leads to more conservative step sizes to compensate.

To estimate $\psi$, one can first performed a dry run with all coefficient set to 0 to estimate $\tau$. All function parameters can be calculated exactly for this problem in terms of the data matrix and $\lambda$. We can then use these parameters and this tau to calculate $\psi$. $\psi$ and $\tau$ merely change the parameters, and do not change execution patterns of the processors. Hence their parameter specification doesn't affect the observed delay. Through simple tuning though, we found that $\psi = 0.25$ resulted in good performance.

In tuning for general problems, there are theoretical reasons why it is difficult to attain acceleration without some prior knowledge of $\sigma$, the strong convexity modulus Arjevani (2017). Ideally $\sigma$ is pre-specified for instance in a regularization term. If the Lipschitz constants $L_i$ cannot be calculated directly (which is rarely the case for the classic dual problem of empirical risk minimization objectives), the line-search method discussed in Roux et al. (2012) Section 4 can be used.

### A.2    Sparse update formulation

As mentioned in Section 5, authors in Lee & Sidford (2013a) proposed a linear transformation of an accelerated RBCD scheme that results in sparse coordinate updates. Our proposed algorithm can be given a similar efficient implementation. We may eliminate $x_k$ from A2BCD, and derive the equivalent iteration below:

$$
\begin{pmatrix} y_{k+1} \\ v_{k+1} \end{pmatrix} = \begin{pmatrix} 1 - \alpha\beta, & \alpha\beta \\ 1 - \beta, & \beta \end{pmatrix} \begin{pmatrix} y_k \\ v_k \end{pmatrix} - \begin{pmatrix} \left( \alpha\sigma^{-1/2} L_{i_k}^{-1/2} + h(1-\alpha) L_{i_k}^{-1} \right) \nabla_{i_k} f(\hat{y}^k) \\ \left( \sigma^{-1/2} L_{i_k}^{-1/2} \right) \nabla_{i_k} f(\hat{y}^k) \end{pmatrix}
$$

$$
\triangleq C \begin{pmatrix} y_k \\ v_k \end{pmatrix} - Q_k
$$

where $C$ and $Q_k$ are defined in the obvious way. Hence we define auxiliary variables $p_k, q_k$ defined via:

$$\begin{pmatrix} y_k \\ v_k \end{pmatrix} = C^k \begin{pmatrix} p_k \\ q_k \end{pmatrix} \tag{A.1}$$

These clearly follow the iteration:

$$\begin{pmatrix} p_{k+1} \\ q_{k+1} \end{pmatrix} = \begin{pmatrix} p_k \\ q_k \end{pmatrix} - C^{-(k+1)} Q_k \tag{A.2}$$

Since the vector $Q_k$ is sparse, we can evolve variables $p_k$, and $q_k$ in a sparse manner, and recover the original iteration variables at the end of the algorithm via A.1.

The gradient of the dual function is given by:

$$\nabla D(y) = \frac{1}{\lambda d} \left( \frac{1}{d} A^T A y + \lambda (y + l) \right)$$

As mentioned before, it is necessary to maintain or recover $A y_k$ to calculate block gradients. Since $A y_k$ can be recovered via the linear relation in equation A.1, and the gradient is an affine function, we maintain the auxiliary vectors $A p^k$ and $A q^k$ instead.

Hence we propose the following efficient implementation in Algorithm 1. We used this to generate the results in Table 5. We also note also that it can improve performance to periodically recover $v_k$ and $y_k$, reset the values of $p^k$, $q^k$, and $C$ to $v_k$, $y_k$, and $I$ respectively, and restarting the scheme (which can be done cheaply in time $\mathcal{O}(d)$).

We let $B \in \mathbb{R}^{2 \times 2}$ represent $C^k$, and $b$ represent $B^{-1}$. $\otimes$ is the Kronecker product. Each computing node has local outdated versions of $p, q, Ap, Aq$ which we denote $\hat{p}, \hat{q}, \hat{A}p, \hat{A}q$ respectively. We also find it convenient to define:

$$\begin{bmatrix} D_1^k \\ D_2^k \end{bmatrix} = \begin{bmatrix} \alpha \sigma^{-1/2} L_{i_k}^{-1/2} + h(1-\alpha) L_{i_k}^{-1} \\ \sigma^{-1/2} L_{i_k}^{-1/2} \end{bmatrix} \tag{A.3}$$

---

**Algorithm 1** Shared-memory implementation of A2BCD

---

1: **Inputs:** Function parameters $A$, $\lambda$, $L$, $\{L_i\}_{i=1}^n$, $n$, $d$. Delay $\tau$ (obtained in dry run). Starting vectors $y$, $v$.
2: **Shared data:** Solution vectors $p$, $q$; auxiliary vectors $Ap$, $Aq$; sparsifying matrix $B$
3: **Node local data:** Solution vectors $\hat{p}$, $\hat{q}$, auxiliary vectors $\hat{A}p$, $\hat{A}q$, sparsifying matrix $\hat{B}$.
4: Calculate parameters $\psi, \alpha, \beta, h$ via 1. Set $k = 0$.
5: **Initializations:** $p \leftarrow y$, $q \leftarrow v$, $Ap \leftarrow Ay$, $Aq \leftarrow Av$, $B \leftarrow I$.
6: **while** not converged, each computing node asynchronous **do**
7:     Randomly select block $i$ via equation 2.1.
8:     Read shared data into local memory: $\hat{p} \leftarrow p$, $\hat{q} \leftarrow q$, $\hat{A}p \leftarrow Ap$, $\hat{A}q \leftarrow Aq$, $\hat{B} \leftarrow B$.
9:     Compute block gradient: $\nabla_i f(\hat{y}) = \frac{1}{n\lambda}\left(\frac{1}{n}A_i^T\left(\hat{B}_{1,1}\hat{A}p + \hat{B}_{1,2}\hat{A}q\right) + \lambda\left(\hat{B}_{1,1}\hat{p} + \hat{B}_{1,2}\hat{q}\right)\right)$
10:     Compute quantity $g_i = A_i^T \nabla_i f(\hat{y})$
    **Shared memory updates:**
11:     Update $B \leftarrow \begin{bmatrix} 1 - \alpha\beta & \alpha\beta \\ 1 - \beta & \beta \end{bmatrix} \times B$, calculate inverse $b \leftarrow B^{-1}$.
12:     $\begin{bmatrix} p \\ q \end{bmatrix} \; -= b \begin{bmatrix} D_1^k \\ D_2^k \end{bmatrix} \otimes \nabla_i f(\hat{y})$ ,    $\begin{bmatrix} Ap \\ Aq \end{bmatrix} \; -= b \begin{bmatrix} D_1^k \\ D_2^k \end{bmatrix} \otimes g_i$
13:     Increase iteration count: $k \leftarrow k + 1$
14: **end while**
15: Recover original iteration variables: $\begin{bmatrix} y \\ v \end{bmatrix} \leftarrow B \begin{bmatrix} p \\ q \end{bmatrix}$. Output $y$.

---

## B  PROOF OF THE MAIN RESULT

We first recall a couple of inequalities for convex functions.

**Lemma 7.** Let $f$ be $\sigma$-strongly convex with $L$-Lipschitz gradient. Then we have:

$$f(y) \leq f(x) + \langle y - x, \nabla f(x)\rangle + \frac{1}{2}L\|y - x\|^2, \; \forall x, y \tag{B.1}$$

$$f(y) \geq f(x) + \langle y - x, \nabla f(x)\rangle + \frac{1}{2}\sigma\|y - x\|^2, \; \forall x, y \tag{B.2}$$

We also find it convenient to define the norm:

$$\|s\|_* = \sqrt{\sum_{i=1}^n L_i^{-1/2}\|s_i\|^2} \tag{B.3}$$

### B.1  Starting point

First notice that using the definition equation 2.8 of $v_{k+1}$ we have:

$$\|v_{k+1}\|^2 = \|\beta v_k + (1-\beta)y_k\|^2 - 2\sigma^{-1/2}L_{i_k}^{-1/2}\langle\beta v_k + (1-\beta)y_k, \nabla_{i_k}f(\hat{y}_k)\rangle + \sigma^{-1}L_{i_k}^{-1}\|\nabla_{i_k}f(\hat{y}_k)\|^2$$

$$\mathbb{E}_k\|v_{k+1}\|^2 = \underbrace{\|\beta v_k + (1-\beta)y_k\|^2}_{A} - 2\sigma^{-1/2}S^{-1}\underbrace{\langle\beta v_k + (1-\beta)y_k, \nabla f(\hat{y}_k)\rangle}_{B} \tag{B.4}$$

$$+ S^{-1}\sigma^{-1}\underbrace{\sum_{i=1}^{n}L_i^{-1/2}\|\nabla_i f(\hat{y}_k)\|^2}_{C}$$

We have the following general identity:

$$\|\beta x + (1-\beta)y\|^2 = \beta\|x\|^2 + (1-\beta)\|y\|^2 - \beta(1-\beta)\|x-y\|^2, \ \forall x, y \tag{B.5}$$

It can also easily be verified from equation 2.6 that we have:

$$v_k = y_k + \alpha^{-1}(1-\alpha)(y_k - x_k) \tag{B.6}$$

Using equation B.5 on term $A$, equation B.6 on term $B$, and recalling the definition equation B.3 on term $C$, we have from equation B.4:

$$\mathbb{E}_k\|v_{k+1}\|^2 = \beta\|v_k\|^2 + (1-\beta)\|y_k\|^2 - \beta(1-\beta)\|v_k - y_k\|^2 + S^{-1}\sigma^{-1/2}\|\nabla f(\hat{y}_k)\|_*^2 \tag{B.7}$$

$$- 2\sigma^{-1/2}S^{-1}\beta\alpha^{-1}(1-\alpha)\langle y_k - x_k, \nabla f(\hat{y}_k)\rangle - 2\sigma^{-1/2}S^{-1}\langle y_k, \nabla f(\hat{y}_k)\rangle$$

This inequality is our starting point. We analyze the terms on the second line in the next section.

### B.2  The Cross Term

To analyze these terms, we need a small lemma. This lemma is fundamental in allowing us to deal with asynchronicity.

**Lemma 8.**  *Let $\chi, A > 0$. Let the delay be bounded by $\tau$. Then:*

$$A\|\hat{y}_k - y_k\| \leq \frac{1}{2}\chi^{-1}A^2 + \frac{1}{2}\chi\tau\sum_{j=1}^{\tau}\|y_{k+1-j} - y_{k-j}\|^2$$

*Proof.* See Hannah & Yin (2017a). $\qquad\qquad\square$

**Lemma 9.**  *We have:*

$$-\langle\nabla f(\hat{y}_k), y_k\rangle \leq -f(y_k) - \frac{1}{2}\sigma(1-\psi)\|y_k\|^2 + \frac{1}{2}\boldsymbol{L\kappa\psi^{-1}\tau\sum_{j=1}^{\tau}\|y_{k+1-j} - y_{k-j}\|^2} \tag{B.8}$$

$$\langle\nabla f(\hat{y}_k), x_k - y_k\rangle \leq f(x_k) - f(y_k) \tag{B.9}$$

$$+ \frac{1}{2}\boldsymbol{L\alpha(1-\alpha)^{-1}\left(\kappa^{-1}\psi\beta\|v_k - y_k\|^2 + \kappa\psi^{-1}\beta^{-1}\tau\sum_{j=1}^{\tau}\|y_{k+1-j} - y_{k-j}\|^2\right)}$$

The terms in bold in equation B.8 and equation B.9 are a result of the asynchronicity, and are identically 0 in its absence.

*Proof.* Our strategy is to separately analyze terms that appear in the traditional analysis of Nesterov (2012), and the terms that result from asynchronicity. We first prove equation B.8:

$$-\langle \nabla f(\hat{y}_k), y_k \rangle = -\langle \nabla f(y_k), y_k \rangle - \langle \nabla f(\hat{y}_k) - \nabla f(y_k), y_k \rangle$$

$$\leq -f(y_k) - \frac{1}{2}\sigma \|y_k\|^2 + L\|\hat{y}_k - y_k\|\|y_k\| \tag{B.10}$$

equation B.10 follows from strong convexity (equation B.2 with $x = y_k$ and $y = x_*$), and the fact that $\nabla f$ is $L$-Lipschitz. The term due to asynchronicity becomes:

$$L\|\hat{y}_k - y_k\|\|y_k\| \leq \frac{1}{2}L\kappa^{-1}\psi\|y_k\|^2 + \frac{1}{2}L\kappa\psi^{-1}\tau\sum_{j=1}^{\tau}\|y_{k+1-j} - y_{k-j}\|^2$$

using Lemma 8 with $\chi = \kappa\psi^{-1}, A = \|y_k\|$. Combining this with equation B.10 completes the proof of equation B.8.

We now prove equation B.9:

$$\langle \nabla f(\hat{y}_k), x_k - y_k \rangle = \langle \nabla f(y_k), x_k - y_k \rangle + \langle \nabla f(\hat{y}_k) - \nabla f(y_k), x_k - y_k \rangle$$

$$\leq f(x_k) - f(y_k) + L\|\hat{y}_k - y_k\|\|x_k - y_k\|$$

$$\leq f(x_k) - f(y_k)$$

$$+ \frac{1}{2}L\left(\kappa^{-1}\psi\beta\alpha^{-1}(1-\alpha)\|x_k - y_k\|^2 + \kappa\psi^{-1}\beta^{-1}\alpha(1-\alpha)^{-1}\tau\sum_{j=1}^{\tau}\|y_{k+1-j} - y_{k-j}\|^2\right)$$

Here the last line follows from Lemma 8 with $\chi = \kappa\psi^{-1}\beta^{-1}\alpha(1-\alpha)^{-1}$, $A = nx_k - y_k$. We can complete the proof using the following identity that can be easily obtained from equation 2.6:

$$y_k - x_k = \alpha(1-\alpha)^{-1}(v_k - y_k) \qquad\qquad \square$$

## B.3   Function-value term

Much like Nesterov (2012), we need a $f(x_k)$ term in the Lyapunov function (see the middle of page 357). However we additionally need to consider asynchronicity when analyzing the growth of this term. Again terms due to asynchronicity are emboldened.

**Lemma 10.**   *We have:*

$$\mathbb{E}_k f(x_{k+1}) \leq f(y_k) - \frac{1}{2}h\left(2 - h\left(1 + \frac{\mathbf{1}}{\mathbf{2}}\boldsymbol{\sigma^{1/2}}\underline{L}^{-1/2}\boldsymbol{\psi}\right)\right)S^{-1}\|\nabla f(\hat{y}_k)\|_*^2$$

$$+ \boldsymbol{S^{-1}L\sigma^{1/2}\kappa\psi^{-1}\tau}\sum_{j=1}^{\tau}\|y_{k+1-j} - y_{k-j}\|^2$$

*Proof.* From the definition equation 2.7 of $x_{k+1}$, we can see that $x_{k+1} - y_k$ is supported on block $i_k$. Since each gradient block $\nabla_i f$ is $L_i$ Lipschitz with respect to changes to block $i$, we can use

equation B.1 to obtain:

$$f(x_{k+1}) \le f(y_k) + \langle \nabla f(y_k), x_{k+1} - y_k \rangle + \frac{1}{2}L_{i_k}\|x_{k+1} - y_k\|^2$$

$$\text{(from equation 2.7)} = f(y_k) - hL_{i_k}^{-1}\langle \nabla_{i_k} f(y_k), \nabla_{i_k} f(\hat{y}_k)\rangle + \frac{1}{2}h^2 L_{i_k}^{-1}\|\nabla_{i_k} f(\hat{y}_k)\|^2$$

$$= f(y_k) - hL_{i_k}^{-1}\langle \nabla_{i_k} f(y_k) - \nabla_{i_k} f(\hat{y}_k), \nabla_{i_k} f(\hat{y}_k)\rangle - \frac{1}{2}h(2-h)L_{i_k}^{-1}\|\nabla_{i_k} f(\hat{y}_k)\|^2$$

$$\mathbb{E}_k f(x_{k+1}) \le f(y_k) - hS^{-1}\sum_{i=1}^n L_i^{-1/2}\langle \nabla_i f(y_k) - \nabla_i f(\hat{y}_k), \nabla_i f(\hat{y}_k)\rangle - \frac{1}{2}h(2-h)S^{-1}\|\nabla f(\hat{y}_k)\|_*^2$$

$$\text{(B.11)}$$

Here the last line followed from the definition equation B.3 of the norm $\|\cdot\|_{*1/2}$. We now analyze the middle term:

$$-\sum_{i=1}^n L_i^{-1/2}\langle \nabla_i f(y_k) - \nabla_i f(\hat{y}_k), \nabla_i f(\hat{y}_k)\rangle$$

$$= -\left\langle \sum_{i=1}^n L_i^{-1/4}(\nabla_i f(y_k) - \nabla_i f(\hat{y}_k)), \sum_{i=1}^n L_i^{-1/4}\nabla_i f(\hat{y}_k)\right\rangle$$

$$\text{(Cauchy Schwarz)} \le \left\|\sum_{i=1}^n L_i^{-1/4}(\nabla_i f(y_k) - \nabla_i f(\hat{y}_k))\right\|\left\|\sum_{i=1}^n L_i^{-1/4}\nabla_i f(\hat{y}_k)\right\|$$

$$= \left(\sum_{i=1}^n L_i^{-1/2}\|\nabla_i f(y_k) - \nabla_i f(\hat{y}_k)\|^2\right)^{1/2}\left(\sum_{i=1}^n L_i^{-1/2}\|\nabla_i f(\hat{y}_k)\|^2\right)^{1/2}$$

$$(\underline{L} \le L_i, \forall i \text{ and equation B.3}) \le \underline{L}^{-1/4}\|\nabla f(y_k) - \nabla f(\hat{y}_k)\|\|\nabla f(\hat{y}_k)\|_*$$

$$(\nabla f \text{ is } L\text{-Lipschitz}) \le \underline{L}^{-1/4}L\|y_k - \hat{y}_k\|\|\nabla f(\hat{y}_k)\|_*$$

We then apply Lemma 8 to this with $\chi = 2h^{-1}\sigma^{1/2}\underline{L}^{1/4}\kappa\psi^{-1}, A = \|\nabla f(\hat{y}_k)\|_*$ to yield:

$$-\sum_{i=1}^n L_i^{-1/2}\langle \nabla_i f(y_k) - \nabla_i f(\hat{y}_k), \nabla_i f(\hat{y}_k)\rangle \le h^{-1}L\sigma^{1/2}\kappa\psi^{-1}\tau\sum_{j=1}^\tau \|y_{k+1-j} - y_{k-j}\|^2 \qquad \text{(B.12)}$$

$$+ \frac{1}{4}h\sigma^{1/2}\underline{L}^{-1/2}\psi\|\nabla f(\hat{y}_k)\|_*^2$$

Finally to complete the proof, we combine equation B.11, with equation B.12. $\qquad\square$

## B.4 ASYNCHRONICITY ERROR

The previous inequalities produced difference terms of the form $\|y_{k+1-j} - y_{k-j}\|^2$. The following lemma shows how these errors can be incorporated into a Lyapunov function.

**Lemma 11.** *Let $0 < r < 1$ and consider the asynchronicity error and corresponding coefficients:*

$$A_k = \sum_{j=1}^\infty c_j\|y_{k+1-j} - y_{k-j}\|^2$$

$$c_i = \sum_{j=i}^\infty r^{i-j-1}s_j$$

*This sum satisfies:*

$$\mathbb{E}_k[A_{k+1} - rA_k] = c_1\mathbb{E}_k\|y_{k+1} - y_k\|^2 - \sum_{j=1}^{\infty} s_j\|y_{k+1-j} - y_{k-j}\|^2$$

**Remark 2. Interpretation.** This result means that an asynchronicity error term $A_k$ can negate a series of difference terms $-\sum_{j=1}^{\infty} s_j\|y_{k+1-j} - y_{k-j}\|^2$ at the cost of producing an additional error $c_1\mathbb{E}_k\|y_{k+1} - y_k\|^2$, while maintaining a convergence rate of $r$. This essentially converts difference terms, which are hard to deal with, into a $\|y_{k+1} - y_k\|^2$ term which can be negated by other terms in the Lyapunov function. The proof is straightforward.

*Proof.*

$$\mathbb{E}_k[A_{k+1} - rA_k] = \mathbb{E}_k\sum_{j=0}^{\infty} c_{j+1}\|y_{k+1-j} - y_{k-j}\|^2 - r\mathbb{E}_k\sum_{j=1}^{\infty} c_j\|y_{k+1-j} - y_{k-j}\|^2$$

$$= c_1\mathbb{E}_k\|y_{k+1} - y_k\|^2 + \mathbb{E}_k\sum_{j=1}^{\infty}(c_{j+1} - rc_j)\|y_{k+1-j} - y_{k-j}\|^2$$

Noting the following completes the proof:

$$c_{i+1} - rc_i = \sum_{j=i+1}^{\infty} r^{i+1-j-1}s_j - r\sum_{j=i}^{\infty} r^{i-j-1}s_j = -s_i \qquad\qquad \square$$

Given that $A_k$ allows us to negate difference terms, we now analyze the cost $c_1\mathbb{E}_k\|y_{k+1} - y_k\|^2$ of this negation.

**Lemma 12.** *We have:*

$$\mathbb{E}_k\|y_{k+1} - y_k\|^2 \leq 2\alpha^2\beta^2\|v_k - y_k\|^2 + 2S^{-1}\underline{L}^{-1}\|\nabla f(\hat{y}_k)\|^2$$

*Proof.*

$$y_{k+1} - y_k = (\alpha v_{k+1} + (1 - \alpha)x_{k+1}) - y_k$$
$$= \alpha\Big(\beta v_k + (1 - \beta)y_k - \sigma^{-1/2}L_{i_k}^{-1/2}\nabla_{i_k} f(\hat{y}_k)\Big) + (1 - \alpha)\big(y_k - hL_{i_k}^{-1}\nabla_{i_k} f(\hat{y}_k)\big) - y_k \tag{B.13}$$

$$= \alpha\beta v_k + \alpha(1 - \beta)y_k - \alpha\sigma^{-1/2}L_{i_k}^{-1/2}\nabla_{i_k} f(\hat{y}_k) - \alpha y_k - (1 - \alpha)hL_{i_k}^{-1}\nabla_{i_k} f(\hat{y}_k)$$

$$= \alpha\beta(v_k - y_k) - \Big(\alpha\sigma^{-1/2}L_{i_k}^{-1/2} + h(1 - \alpha)L_{i_k}^{-1}\Big)\nabla_{i_k} f(\hat{y}_k)$$

$$\|y_{k+1} - y_k\|^2 \leq 2\alpha^2\beta^2\|v_k - y_k\|^2 + 2\Big(\alpha\sigma^{-1/2}L_{i_k}^{-1/2} + h(1 - \alpha)L_{i_k}^{-1}\Big)^2\|\nabla_{i_k} f(\hat{y}_k)\|^2 \tag{B.14}$$

Here equation B.13 following from equation 2.8, the definition of $v_{k+1}$. equation B.14 follows from the inequality $\|x + y\|^2 \le 2\|x\|^2 + 2\|y\|^2$. The rest is simple algebraic manipulation.

$$\|y_{k+1} - y_k\|^2 \le 2\alpha^2\beta^2\|v_k - y_k\|^2 + 2L_{i_k}^{-1}\Big(\alpha\sigma^{-1/2} + h(1-\alpha)L_{i_k}^{-1/2}\Big)^2\|\nabla_{i_k}f(\hat{y}_k)\|^2$$

$$(\underline{\mathrm{L}} \le L_i, \forall i) \le 2\alpha^2\beta^2\|v_k - y_k\|^2 + 2L_{i_k}^{-1}\Big(\alpha\sigma^{-1/2} + h(1-\alpha)\underline{\mathrm{L}}^{-1/2}\Big)^2\|\nabla_{i_k}f(\hat{y}_k)\|^2$$

$$= 2\alpha^2\beta^2\|v_k - y_k\|^2 + 2L_{i_k}^{-1}\underline{\mathrm{L}}^{-1}\Big(\underline{\mathrm{L}}^{1/2}\sigma^{-1/2}\alpha + h(1-\alpha)\Big)^2\|\nabla_{i_k}f(\hat{y}_k)\|^2$$

$$\mathbb{E}\|y_{k+1} - y_k\|^2 \le 2\alpha^2\beta^2\|v_k - y_k\|^2 + 2S^{-1}\underline{\mathrm{L}}^{-1}\Big(\underline{\mathrm{L}}^{1/2}\sigma^{-1/2}\alpha + h(1-\alpha)\Big)^2\|\nabla f(\hat{y}_k)\|_*^2$$

Finally, to complete the proof, we prove $\underline{\mathrm{L}}^{1/2}\sigma^{-1/2}\alpha + h(1-\alpha) \le 1$.

$$\underline{\mathrm{L}}^{1/2}\sigma^{-1/2}\alpha + h(1-\alpha) = h + \alpha\Big(\underline{\mathrm{L}}^{1/2}\sigma^{-1/2} - h\Big)$$

(definitions of $h$ and $\alpha$: equation 2.3, and equation 2.5) $= 1 - \dfrac{1}{2}\sigma^{1/2}\underline{\mathrm{L}}^{-1/2}\psi + \sigma^{1/2}S^{-1}\Big(\underline{\mathrm{L}}^{1/2}\sigma^{-1/2}\Big)$

$$\le 1 - \sigma^{1/2}\underline{\mathrm{L}}^{-1/2}\left(\frac{1}{2}\psi - \sigma^{-1/2}S^{-1}\underline{\mathrm{L}}^1\right) \tag{B.15}$$

Rearranging the definition of $\psi$, we have:

$$S^{-1} = \frac{1}{9^2}\psi^2\underline{\mathrm{L}}^1 L^{-3/2}\kappa^{-1/2}\tau^{-2}$$

$$(\tau \ge 1 \text{ and } \psi \le \frac{1}{2}) \le \frac{1}{18^2}\underline{\mathrm{L}}^1 L^{-3/2}\kappa^{-1/2}$$

Using this on equation B.15, we have:

$$\underline{\mathrm{L}}^{1/2}\alpha\sigma^{-1/2} + h(1-\alpha) \le 1 - \sigma^{1/2}\underline{\mathrm{L}}^{-1/2}\left(\frac{1}{2}\psi - \frac{1}{18^2}\underline{\mathrm{L}}^1 L^{-3/2}\kappa^{-1/2}\sigma^{-1/2}\underline{\mathrm{L}}^1\right)$$

$$= 1 - \sigma^{1/2}\underline{\mathrm{L}}^{-1/2}\left(\frac{1}{2}\psi - \frac{1}{18^2}(\underline{\mathrm{L}}/L)^2\right)$$

$$(\psi \le \frac{1}{2}) = 1 - \sigma^{1/2}\underline{\mathrm{L}}^{-1/2}\left(\frac{1}{24} - \frac{1}{18^2}\right) \le 1.$$

This completes the proof. □

## B.5   Master inequality

We are finally in a position to bring together all the all the previous results together into a master inequality for the Lyapunov function $\rho_k$ (defined in equation 2.11). After this lemma is proven, we will prove that the right hand size is negative, which will imply that $\rho_k$ linearly converges to 0 with rate $\beta$.

**Lemma 13. Master inequality.**  *We have:*

$$\mathbb{E}_k[\rho_{k+1} - \beta\rho_k]$$

$$\leq + \|y_k\|^2 \qquad\qquad\qquad \times\left(1 - \beta - \sigma^{-1/2}S^{-1}\sigma(1-\psi)\right) \tag{B.16}$$

$$+ \|v_k - y_k\|^2 \qquad\qquad \times\beta\left(2\alpha^2\beta c_1 + S^{-1}\beta L^{1/2}\kappa^{-1/2}\psi - (1-\beta)\right)$$

$$+ f(y_k) \qquad\qquad\qquad \times\left(c - 2\sigma^{-1/2}S^{-1}\big(\beta\alpha^{-1}(1-\alpha) + 1\big)\right)$$

$$+ f(x_k) \qquad\qquad\qquad \times\beta\left(2\sigma^{-1/2}S^{-1}\alpha^{-1}(1-\alpha) - c\right)$$

$$+ \sum_{j=1}^{\tau}\|y_{k+1-j} - y_{k-j}\|^2 \quad \times S^{-1}L\kappa\psi^{-1}\tau\sigma^{1/2}\big(2\sigma^{-1} + c\big) - s$$

$$+ \|\nabla f(\hat{y}_k)\|_*^2 \qquad\qquad \times S^{-1}\left(\sigma^{-1} + 2\underline{L}^{-1}c_1 - \frac{1}{2}ch\left(2 - h\left(1 + \frac{1}{2}\sigma^{1/2}\underline{L}^{-1/2}\psi\right)\right)\right)$$

*Proof.*

$$\mathbb{E}_k\|v_{k+1}\|^2 - \beta\|v_k\|^2$$

$$(\text{B.7}) = (1-\beta)\|y_k\|^2 - \beta(1-\beta)\|v_k - y_k\|^2 + S^{-1}\sigma^{-1}\|\nabla f(\hat{y}_k)\|_*^2$$

$$- 2\sigma^{-1/2}S^{-1}\langle y_k, \nabla f(\hat{y}_k)\rangle$$

$$- 2\sigma^{-1/2}S^{-1}\beta\alpha^{-1}(1-\alpha)\langle y_k - x_k, \nabla f(\hat{y}_k)\rangle$$

$$\leq (1-\beta)\|y_k\|^2 - \beta(1-\beta)\|v_k - y_k\|^2 + S^{-1}\sigma^{-1}\|\nabla f(\hat{y}_k)\|_*^2 \tag{B.17}$$

$$(\text{B.8}) + 2\sigma^{-1/2}S^{-1}\left(-f(y_k) - \frac{1}{2}\sigma(1-\psi)\|y_k\|^2 + \frac{1}{2}L\kappa\psi^{-1}\tau\sum_{j=1}^{\tau}\|y_{k+1-j} - y_{k-j}\|^2\right)$$

$$(\text{B.9}) - 2\sigma^{-1/2}S^{-1}\beta\alpha^{-1}(1-\alpha)(f(x_k) - f(y_k))$$

$$+ \sigma^{-1/2}S^{-1}\beta L\left(\kappa^{-1}\psi\beta\|v_k - y_k\|^2 + \kappa\psi^{-1}\beta^{-1}\tau\sum_{j=1}^{\tau}\|y_{k+1-j} - y_{k-j}\|^2\right)$$

We now collect and organize the similar terms of this inequality.

$$\leq + \|y_k\|^2 \qquad\qquad\qquad \times\left(1 - \beta - \sigma^{-1/2}S^{-1}\sigma(1-\psi)\right)$$

$$+ \|v_k - y_k\|^2 \qquad\qquad \times\beta\left(\sigma^{-1/2}S^{-1}\beta L\kappa^{-1}\psi - (1-\beta)\right)$$

$$- f(y_k) \qquad\qquad\qquad \times 2\sigma^{-1/2}S^{-1}\big(\beta\alpha^{-1}(1-\alpha) + 1\big)$$

$$+ f(x_k) \qquad\qquad\qquad \times 2\sigma^{-1/2}S^{-1}\beta\alpha^{-1}(1-\alpha)$$

$$+ \sum_{j=1}^{\tau}\|y_{k+1-j} - y_{k-j}\|^2 \quad \times 2\sigma^{-1/2}S^{-1}L\kappa\psi^{-1}\tau$$

$$+ \|\nabla f(\hat{y}_k)\|_*^2 \qquad\qquad \times\sigma^{-1}S^{-1}$$

Now finally, we add the function-value and asynchronicity terms to our analysis. We use Lemma 11 is with $r = 1 - \sigma^{1/2} S^{-1}$, and

$$s_i = \begin{cases} s = 6 S^{-1} L^{1/2} \kappa^{3/2} \psi^{-1} \tau, & 1 \le i \le \tau \\ 0, & i > \tau \end{cases} \tag{B.18}$$

Notice that this choice of $s_i$ will recover the coefficient formula given in equation 2.9. Hence we have:

$$\mathbb{E}_k[cf(x_{k+1}) + A_{k+1} - \beta(cf(x_k) + A_k)]$$

$$\text{(Lemma 10)} \le cf(y_k) - \frac{1}{2} ch\left(2 - h\left(1 + \frac{1}{2}\sigma^{1/2}\underline{\mathrm{L}}^{-1/2}\psi\right)\right)S^{-1}\|\nabla f(\hat{y}_k)\|_*^2 - \beta cf(x_k) \tag{B.19}$$

$$+ S^{-1} L \sigma^{1/2} \kappa \psi^{-1} \tau \sum_{j=1}^{\tau} \|y_{k+1-j} - y_{k-j}\|^2$$

$$\text{(Lemmas 11 and 12)} + c_1\left(2\alpha^2\beta^2\|v_k - y_k\|^2 + 2S^{-1}\underline{\mathrm{L}}^{-1}\|\nabla f(\hat{y}_k)\|^2\right) \tag{B.20}$$

$$- \sum_{j=1}^{\infty} s_j \|y_{k+1-j} - y_{k-j}\|^2 + A_k(r - \beta)$$

Notice $A_k(r - \beta) \le 0$. Finally, combining equation B.17 and equation B.19 completes the proof. $\qquad\square$

In the next section, we will prove that every coefficient on the right hand side of equation B.16 is 0 or less, which will complete the proof of Theorem 1.

## B.6    Proof of main theorem

**Lemma 14.**    The coefficients of $\|y_k\|^2$, $f(y_k)$, and $\sum_{j=1}^{\tau}\|y_{k+1-j} - y_{k-j}\|^2$ in Lemma 13 are non-positive.

*Proof.* The coefficient $1 - (1 - \psi)\sigma^{1/2}S^{-1} - \beta$ of $\|y_k\|^2$ is identically 0 via the definition equation 2.4 of $\beta$. The coefficient $c - 2\sigma^{-1/2}S^{-1}\big(\beta\alpha^{-1}(1 - \alpha) + 1\big)$ of $f(y_k)$ is identically 0 via the definition equation 2.12 of $c$.

First notice from the definition equation 2.12 of $c$:

$$c = 2\sigma^{-1/2}S^{-1}\big(\beta\alpha^{-1}(1 - \alpha) + 1\big)$$

$$\text{(definitions of } \alpha, \beta) = 2\sigma^{-1/2}S^{-1}\Big(\big(1 - \sigma^{1/2}S^{-1}(1 - \psi)\big)(1 + \psi)\sigma^{-1/2}S + 1\Big)$$

$$= 2\sigma^{-1/2}S^{-1}\Big((1 + \psi)\sigma^{-1/2}S + \psi^2\Big)$$

$$= 2\sigma^{-1}\Big((1 + \psi) + \psi^2\sigma^{1/2}S^{-1}\Big) \tag{B.21}$$

$$c \le 4\sigma^{-1} \tag{B.22}$$

Here the last line followed since $\psi \leq \frac{1}{2}$ and $\sigma^{1/2}S^{-1} \leq 1$. We now analyze the coefficient of $\sum_{j=1}^{\tau} \|y_{k+1-j} - y_{k-j}\|^2$.

$$S^{-1}L\kappa\psi^{-1}\tau\sigma^{1/2}(2\sigma^{-1} + c) - s$$
$$\text{(B.22)} \leq 6L^{1/2}\kappa^{3/2}\psi^{-1}\tau - s$$
$$\text{(definition equation B.18 of } s) \leq 0 \qquad\qquad \square$$

**Lemma 15.** The coefficient $\beta(2\sigma^{-1/2}S^{-1}\alpha^{-1}(1-\alpha) - c)$ of $f(x_k)$ in Lemma 13 is non-positive.

*Proof.*

$$2\sigma^{-1/2}S^{-1}\alpha^{-1}(1-\alpha) - c$$
$$\text{(B.21)} = 2\sigma^{-1/2}S^{-1}(1+\psi)\sigma^{-1/2}S - 2\sigma^{-1}\left((1+\psi) + \psi^2\sigma^{1/2}S^{-1}\right)$$
$$= 2\sigma^{-1}\left((1+\psi) - \left((1+\psi) + \psi^2\sigma^{1/2}S^{-1}\right)\right)$$
$$= -2\psi^2\sigma^{-1/2}S^{-1} \leq 0 \qquad\qquad \square$$

**Lemma 16.** The coefficient $S^{-1}\left(\sigma^{-1} + 2\underline{L}^{-1}c_1 - \frac{1}{2}ch\left(2 - h\left(1 + \frac{1}{2}\sigma^{1/2}\underline{L}^{-1/2}\psi\right)\right)\right)$ of $\|\nabla f(\hat{y}_k)\|_*^2$ in Lemma 13 is non-positive.

*Proof.* We first need to bound $c_1$.

$$\text{(equation B.18 and equation 2.9)} \quad c_1 = s\sum_{j=1}^{\tau}\left(1 - \sigma^{1/2}S^{-1}\right)^{-j}$$
$$\text{equation B.18} \quad \leq 6S^{-1}L^{1/2}\kappa^{3/2}\psi^{-1}\tau\sum_{j=1}^{\tau}\left(1 - \sigma^{1/2}S^{-1}\right)^{-j}$$
$$\leq 6S^{-1}L^{1/2}\kappa^{3/2}\psi^{-1}\tau^2\left(1 - \sigma^{1/2}S^{-1}\right)^{-\tau}$$

It can be easily verified that if $x \leq \frac{1}{2}$ and $y \geq 0$, then $(1-x)^{-y} \leq \exp(2xy)$. Using this fact with $x = \sigma^{1/2}S^{-1}$ and $y = \tau$, we have:

$$\leq 6S^{-1}L^{1/2}\kappa^{3/2}\psi^{-1}\tau^2\exp\left(\tau\sigma^{1/2}S^{-1}\right)$$
$$\text{(since } \psi \leq 3/7 \text{ and hence } \tau\sigma^{1/2}S^{-1} \leq \frac{1}{7}) \leq S^{-1}L^{1/2}\kappa^{3/2}\psi^{-1}\tau^2 \times 6\exp\left(\frac{1}{7}\right)$$
$$c_1 \leq 7S^{-1}L^{1/2}\kappa^{3/2}\psi^{-1}\tau^2 \qquad\qquad \text{(B.23)}$$

We now analyze the coefficient of $\|\nabla f(\hat{y}_k)\|_*^2$

$$\sigma^{-1} + 2\underline{L}^{-1}c_1 - \frac{1}{2}ch\left(2 - h\left(1 + \frac{1}{2}\sigma^{1/2}\underline{L}^{-1/2}\psi\right)\right)$$

$$(\text{B.23 and 2.5}) \leq \sigma^{-1} + 14S^{-1}\underline{L}^{-1}L^{1/2}\kappa^{3/2}\psi^{-1}\tau^2 - \frac{1}{2}ch\left(1 + \frac{1}{4}\sigma^1\underline{L}^{-1}\psi^2\right)$$

$$\leq \sigma^{-1} + 14S^{-1}\underline{L}^{-1}L^{1/2}\kappa^{3/2}\psi^{-1}\tau^2 - \frac{1}{2}ch$$

$$(\text{definition 2.2 of } \psi) = \sigma^{-1} + \frac{14}{81}\sigma^{-1}\psi - \frac{1}{2}ch$$

$$(\text{B.21, definition 2.5 of } h) = \sigma^{-1}\left(1 + \frac{14}{81}\psi - \left((1 + \psi) + \psi^2\sigma^{1/2}S^{-1}\right)\left(1 - \frac{1}{2}\sigma^{1/2}\underline{L}^{-1/2}\psi\right)\right)$$

$$(\sigma^{1/2}\underline{L}^{-1/2} \leq 0 \text{ and } \sigma^{1/2}S^{-1} \leq 1) \leq \sigma^{-1}\left(1 + \frac{14}{81}\psi - (1 + \psi)\left(1 - \frac{1}{2}\psi\right)\right)$$

$$= \sigma^{-1}\psi\left(\frac{14}{81} + \frac{1}{2}\psi - \frac{1}{2}\right)$$

$$(\psi \leq \frac{1}{2}) \leq 0 \qquad\qquad \square$$

**Lemma 17.** The coefficient $\beta\left(2\alpha^2\beta c_1 + S^{-1}\beta L^{1/2}\kappa^{-1/2}\psi - (1 - \beta)\right)$ of $\|v_k - y_k\|^2$ in 13 is non-positive.

*Proof.*

$$2\alpha^2\beta c_1 + \sigma^{1/2}S^{-1}\beta\psi - (1 - \psi)\sigma^{1/2}S^{-1}$$

$$(\text{B.23}) \leq 14\alpha^2\beta S^{-1}L^{1/2}\kappa^{3/2}\psi^{-1}\tau^2 + \sigma^{1/2}S^{-1}\beta\psi - (1 - \psi)\sigma^{1/2}S^{-1}$$

$$\leq 14\sigma S^{-3}L^{1/2}\kappa^{3/2}\psi^{-1}\tau^2 + \sigma^{1/2}S^{-1}\psi - (1 - \psi)\sigma^{1/2}S^{-1}$$

$$= \sigma^{1/2}S^{-1}\left(14S^{-2}L\kappa\tau^2\psi^{-1} + 2\psi - 1\right)$$

Here the last inequality follows since $\beta \leq 1$ and $\alpha \leq \sigma^{1/2}S^{-1}$. We now rearrange the definition of $\psi$ to yield the identity:

$$S^{-2}\kappa = \frac{1}{9^4}\underline{L}^2L^{-3}\tau^{-4}\psi^4$$

Using this, we have:

$$14S^{-2}L\kappa\tau^2\psi^{-1} + 2\psi - 1$$

$$= \frac{14}{9^4}\underline{L}^2L^{-2}\psi^3\tau^{-2} + 2\psi - 1$$

$$\leq \frac{14}{9^4}\left(\frac{3}{7}\right)^3 1^{-2} + \frac{6}{7} - 1 \leq 0$$

Here the last line followed since $\underline{L} \leq L$, $\psi \leq \frac{3}{7}$, and $\tau \geq 1$. Hence the proof is complete. $\qquad \square$

*Proof of Theorem 1.* Using the master inequality 13 in combination with the previous Lemmas 14, 15, 16, and 17, we have:

$$\mathbb{E}_k[\rho_{k+1}] \leq \beta\rho_k = \left(1 - (1 - \psi)\sigma^{1/2}S^{-1}\right)\rho_k$$

When we have:

$$\left(1 - (1 - \psi)\sigma^{1/2}S^{-1}\right)^k \le \epsilon$$

then the Lyapunov function $\rho_k$ has decreased below $\epsilon\rho_0$ in expectation. Hence the complexity $K(\epsilon)$ satisfies:

$$K(\epsilon)\ln\left(1 - (1 - \psi)\sigma^{1/2}S^{-1}\right) = \ln(\epsilon)$$

$$K(\epsilon) = \frac{-1}{\ln\left(1 - (1 - \psi)\sigma^{1/2}S^{-1}\right)}\ln(1/\epsilon)$$

Now it can be shown that for $0 < x \le \frac{1}{2}$, we have:

$$\frac{1}{x} - 1 \le \frac{-1}{\ln(1 - x)} \le \frac{1}{x} - \frac{1}{2}$$

$$\frac{-1}{\ln(1 - x)} = \frac{1}{x} + \mathcal{O}(1)$$

Since $n \ge 2$, we have $\sigma^{1/2}S^{-1} \le \frac{1}{2}$. Hence:

$$K(\epsilon) = \frac{1}{1 - \psi}\left(\sigma^{-1/2}S + \mathcal{O}(1)\right)\ln(1/\epsilon)$$

An expression for $K_{\texttt{NU\_ACDM}}(\epsilon)$, the complexity of $\texttt{NU\_ACDM}$ follows by similar reasoning.

$$K_{\texttt{NU\_ACDM}}(\epsilon) = \left(\sigma^{-1/2}S + \mathcal{O}(1)\right)\ln(1/\epsilon) \tag{B.24}$$

Finally we have:

$$K(\epsilon) = \frac{1}{1 - \psi}\left(\frac{\sigma^{-1/2}S + \mathcal{O}(1)}{\sigma^{-1/2}S + \mathcal{O}(1)}\right)K_{\texttt{NU\_ACDM}}(\epsilon)$$

$$= \frac{1}{1 - \psi}(1 + o(1))K_{\texttt{NU\_ACDM}}(\epsilon)$$

which completes the proof. $\qquad\square$

## C    Ordinary Differential Equation Analysis

### C.1    Derivation of ODE for synchronous A2BCD

If we take expectations with respect to $\mathbb{E}_k$, then synchronous (no delay) $\texttt{A2BCD}$ becomes:

$$y_k = \alpha v_k + (1 - \alpha)x_k$$
$$\mathbb{E}_k x_{k+1} = y_k - n^{-1}\kappa^{-1}\nabla f(y_k)$$
$$\mathbb{E}_k v_{k+1} = \beta v_k + (1 - \beta)y_k - n^{-1}\kappa^{-1/2}\nabla f(y_k)$$

We find it convenient to define $\eta = n\kappa^{1/2}$. Inspired by this, we consider the following iteration:

$$y_k = \alpha v_k + (1 - \alpha)x_k \tag{C.1}$$

$$x_{k+1} = y_k - s^{1/2}\kappa^{-1/2}\eta^{-1}\nabla f(y_k) \tag{C.2}$$

$$v_{k+1} = \beta v_k + (1 - \beta)y_k - s^{1/2}\eta^{-1}\nabla f(y_k) \tag{C.3}$$

for coefficients:

$$\alpha = \left(1 + s^{-1/2}\eta\right)^{-1}$$

$$\beta = 1 - s^{1/2}\eta^{-1}$$

$s$ is a discretization scale parameter that will be sent to 0 to obtain an ODE analogue of synchronous A2BCD. We first use equation B.6 to eliminate $v_k$ from from equation C.3.

$$0 = -v_{k+1} + \beta v_k + (1 - \beta)y_k - s^{1/2}\eta^{-1}\nabla f(y_k)$$

$$0 = -\alpha^{-1}y_{k+1} + \alpha^{-1}(1 - \alpha)x_{k+1}$$
$$+ \beta\left(\alpha^{-1}y_k - \alpha^{-1}(1 - \alpha)x_k\right) + (1 - \beta)y_k - s^{1/2}\eta^{-1}\nabla f(y_k)$$

$$\text{(times by } \alpha\text{)} \quad 0 = -y_{k+1} + (1 - \alpha)x_{k+1}$$
$$+ \beta(y_k - (1 - \alpha)x_k) + \alpha(1 - \beta)y_k - \alpha s^{1/2}\eta^{-1}\nabla f(y_k)$$
$$= -y_{k+1} + y_k(\beta + \alpha(1 - \beta))$$
$$+ (1 - \alpha)x_{k+1} - x_k\beta(1 - \alpha) - \alpha s^{1/2}\eta^{-1}\nabla f(y_k)$$

We now eliminate $x_k$ using equation C.1:

$$0 = -y_{k+1} + y_k(\beta + \alpha(1 - \beta))$$
$$+ (1 - \alpha)\left(y_k - s^{1/2}\eta^{-1}\kappa^{-1/2}\nabla f(y_k)\right) - \left(y_{k-1} - s^{1/2}\eta^{-1}\kappa^{-1/2}\nabla f(y_{k-1})\right)\beta(1 - \alpha)$$
$$- \alpha s^{1/2}\eta^{-1}\nabla f(y_k)$$
$$= -y_{k+1} + y_k(\beta + \alpha(1 - \beta) + (1 - \alpha)) - \beta(1 - \alpha)y_{k-1}$$
$$+ s^{1/2}\eta^{-1}\nabla f(y_{k-1})(\beta - 1)(1 - \alpha)$$
$$- \alpha s^{1/2}\eta^{-1}\nabla f(y_k)$$
$$= (y_k - y_{k+1}) + \beta(1 - \alpha)(y_k - y_{k-1})$$
$$+ s^{1/2}\eta^{-1}(\nabla f(y_{k-1})(\beta - 1)(1 - \alpha) - \alpha\nabla f(y_k))$$

Now to derive an ODE, we let $y_k = Y\left(ks^{1/2}\right)$. Then $\nabla f(y_{k-1}) = \nabla f(y_k) + \mathcal{O}\left(s^{1/2}\right)$. Hence the above becomes:

$$0 = (y_k - y_{k+1}) + \beta(1 - \alpha)(y_k - y_{k-1})$$
$$+ s^{1/2}\eta^{-1}((\beta - 1)(1 - \alpha) - \alpha)\nabla f(y_k) + \mathcal{O}\left(s^{3/2}\right)$$
$$0 = \left(-s^{1/2}\dot{Y} - \frac{1}{2}s\ddot{Y}\right) + \beta(1 - \alpha)\left(s^{1/2}\dot{Y} - \frac{1}{2}s\ddot{Y}\right) \tag{C.4}$$
$$+ s^{1/2}\eta^{-1}((\beta - 1)(1 - \alpha) - \alpha)\nabla f(y_k) + \mathcal{O}\left(s^{3/2}\right)$$

We now look at some of the terms in this equation to find the highest-order dependence on $s$.

$$\beta(1-\alpha) = \left(1 - s^{1/2}\eta^{-1}\right)\left(1 - \frac{1}{1 + s^{-1/2}\eta}\right)$$

$$= \left(1 - s^{1/2}\eta^{-1}\right)\frac{s^{-1/2}\eta}{1 + s^{-1/2}\eta}$$

$$= \frac{s^{-1/2}\eta - 1}{s^{-1/2}\eta + 1}$$

$$= \frac{1 - s^{1/2}\eta^{-1}}{1 + s^{1/2}\eta^{-1}}$$

$$= 1 - 2s^{1/2}\eta^{-1} + \mathcal{O}(s)$$

We also have:

$$(\beta - 1)(1 - \alpha) - \alpha = \beta(1-\alpha) - 1$$

$$= -2s^{1/2}\eta^{-1} + \mathcal{O}(s)$$

Hence using these facts on equation C.4, we have:

$$0 = \left(-s^{1/2}\dot{Y} - \frac{1}{2}s\ddot{Y}\right) + \left(1 - 2s^{1/2}\eta^{-1} + \mathcal{O}(s)\right)\left(s^{1/2}\dot{Y} - \frac{1}{2}s\ddot{Y}\right)$$

$$+ s^{1/2}\eta^{-1}\left(-2s^{1/2}\eta^{-1} + \mathcal{O}(s)\right)\nabla f(y_k) + \mathcal{O}\left(s^{3/2}\right)$$

$$0 = -s^{1/2}\dot{Y} - \frac{1}{2}s\ddot{Y} + \left(s^{1/2}\dot{Y} - \frac{1}{2}s\ddot{Y} - 2s^1\eta^{-1}\dot{Y} + \mathcal{O}\left(s^{3/2}\right)\right)$$

$$\left(-2s^1\eta^{-2} + \mathcal{O}\left(s^{3/2}\right)\right)\nabla f(y_k) + \mathcal{O}\left(s^{3/2}\right)$$

$$0 = -s\ddot{Y} - 2s\eta^{-1}\dot{Y} - 2s\eta^{-2}\nabla f(y_k) + \mathcal{O}\left(s^{3/2}\right)$$

$$0 = -\ddot{Y} - 2\eta^{-1}\dot{Y} - 2\eta^{-2}\nabla f(y_k) + \mathcal{O}\left(s^{1/2}\right)$$

Taking the limit as $s \to 0$, we obtain the ODE:

$$\ddot{Y}(t) + 2\eta^{-1}\dot{Y} + 2\eta^{-2}\nabla f(Y) = 0$$

## C.2   CONVERGENCE PROOF FOR SYNCHRONOUS ODE

$$e^{-\eta^{-1}t}E'(t) = \left\langle\nabla f(Y(t)), \dot{Y}(t)\right\rangle + \eta^{-1}f(Y(t))$$

$$+ \frac{1}{2}\left\langle Y(t) + \eta\dot{Y}(t), \dot{Y}(t) + \eta\ddot{Y}(t)\right\rangle + \eta^{-1}\frac{1}{4}\left\|Y(t) + \eta\dot{Y}(t)\right\|^2$$

(strong convexity equation B.2) $\leq \left\langle\nabla f(Y), \dot{Y}\right\rangle + \eta^{-1}\left\langle\nabla f(Y), Y\right\rangle - \frac{1}{2}\eta^{-1}\|Y\|^2$

$$+ \frac{1}{2}\left\langle Y + \eta\dot{Y}, -\dot{Y} - 2\eta^{-1}\nabla f(Y)\right\rangle + \eta^{-1}\frac{1}{4}\left\|Y(t) + \eta\dot{Y}(t)\right\|^2$$

$$= -\frac{1}{4}\eta^{-1}\|Y\|^2 - \frac{1}{4}\eta\|\dot{Y}\|^2 \leq 0$$

Hence we have $E'(t) \leq 0$. Therefore $E(t) \leq E(0)$. That is:

$$E(t) = e^{n^{-1}\kappa^{-1/2}t}\left(f(Y) + \frac{1}{4}\|Y + \eta\dot{Y}\|^2\right) \leq E(0) = f(Y(0)) + \frac{1}{4}\|Y(0) + \eta\dot{Y}(0)\|^2 \qquad \text{(C.5)}$$

which implies:

$$f(Y(t)) + \frac{1}{4}\|Y(t) + \eta\dot{Y}(t)\|^2 \leq e^{-n^{-1}\kappa^{-1/2}t}\left(f(Y(0)) + \frac{1}{4}\|Y(0) + \eta\dot{Y}(0)\|^2\right) \qquad \text{(C.6)}$$

## C.3 Asynchronicity error lemma

This result is the continuous-time analogue of Lemma 11. First notice that $c(0) = c_0$ and $c(\tau) = 0$. We also have:

$$c'(t)/c_0 = -re^{-rt} - re^{-rt}\frac{e^{-r\tau}}{1 - e^{-r\tau}}$$

$$= -r\left(e^{-rt} + e^{-rt}\frac{e^{-r\tau}}{1 - e^{-r\tau}}\right)$$

$$= -r\left(e^{-rt} + (e^{-rt} - 1)\frac{e^{-r\tau}}{1 - e^{-r\tau}} + \frac{e^{-r\tau}}{1 - e^{-r\tau}}\right)$$

$$c'(t) = -rc(t) - rc_0\frac{e^{-r\tau}}{1 - e^{-r\tau}}$$

Hence using $c(\tau) = 0$:

$$A'(t) = c_0\|\dot{Y}(t)\|^2 + \int_{t-\tau}^{t} c'(t-s)\|\dot{Y}(s)\|^2 ds$$

$$= c_0\|\dot{Y}(t)\|^2 - rA(t) - rc_0\frac{e^{-r\tau}}{1 - e^{-r\tau}}D(t)$$

Now when $x \leq \frac{1}{2}$, we have $\frac{e^{-x}}{1-e^{-x}} \geq \frac{1}{2}x^{-1}$. Hence when $r\tau \leq \frac{1}{2}$, we have:

$$A'(t) \leq c_0\|\dot{Y}(t)\|^2 - rA(t) - \frac{1}{2}\tau^{-1}c_0 D(t)$$

and the result easily follows.

## C.4 Convergence analysis for the asynchronous ODE

We consider the same energy as in the synchronous case (that is, the ODE in equation 3.1). Similar to before, we have:

$$e^{-\eta^{-1}t}E'(t) \leq \langle \nabla f(Y), \dot{Y} \rangle + \eta^{-1}\langle \nabla f(Y), Y \rangle - \frac{1}{2}\eta^{-1}\|Y\|^2$$

$$+ \frac{1}{2}\left\langle Y + \eta\dot{Y}, -\dot{Y} - 2\eta^{-1}\nabla f(\hat{Y})\right\rangle + \eta^{-1}\frac{1}{4}\|Y(t) + \eta\dot{Y}(t)\|^2$$

$$= \langle \nabla f(Y), \dot{Y} \rangle + \eta^{-1}\langle \nabla f(Y), Y \rangle - \frac{1}{2}\eta^{-1}\|Y\|^2$$

$$+ \frac{1}{2}\langle Y + \eta\dot{Y}, -\dot{Y} - 2\eta^{-1}\nabla f(Y) \rangle + \eta^{-1}\frac{1}{4}\|Y(t) + \eta\dot{Y}(t)\|^2$$

$$- \eta^{-1}\left\langle Y + \eta\dot{Y}, \nabla f(\hat{Y}) - \nabla f(Y)\right\rangle$$

$$= -\frac{1}{4}\eta^{-1}\|Y\|^2 - \frac{1}{4}\eta\|\dot{Y}\|^2 - \eta^{-1}\left\langle Y + \eta\dot{Y}, \nabla f(\hat{Y}) - \nabla f(Y)\right\rangle$$

where the final equality follows from the proof in Section C.2. Hence

$$e^{-\eta^{-1}t}E'(t) \leq -\frac{1}{4}\eta^{-1}\|Y\|^2 - \frac{1}{4}\eta\|\dot{Y}\|^2 + L\eta^{-1}\|Y\|\left\|\hat{Y} - Y\right\| + L\|\dot{Y}\|\left\|\hat{Y} - Y\right\| \tag{C.7}$$

Now we present an inequality that is similar to equation 8.

**Lemma 18.** *Let $A, \chi > 0$. Then:*

$$\left\|Y(t) - \hat{Y}(t)\right\|A \leq \frac{1}{2}\chi\tau D(t) + \frac{1}{2}\chi^{-1}A^2$$

*Proof.* Since $\hat{Y}(t)$ is a delayed version of $Y(t)$, we have: $\hat{Y}(t) = Y(t - j(t))$ for some function $j(t) \geq 0$ (though this can be easily generalized to an inconsistent read). Recall that for $\chi > 0$, we have $ab \leq \frac{1}{2}(\chi a^2 + \chi^{-1}b^2)$. Hence

$$X(t) - \hat{X}(t) = \int_{s=t-j(t)}^{t} X'(s)ds$$

$$\left\|X(t) - \hat{X}(t)\right\|A = \left\|\int_{s=t-j(t)}^{t} X'(s)ds\right\|A$$

$$\leq \frac{1}{2}\chi\left\|\int_{s=t-j(t)}^{t} X'(s)ds\right\|^2 + \frac{1}{2}\chi^{-1}A^2$$

$$\text{(Holder's inequality)} \leq \frac{1}{2}\chi\left(\int_{s=t-j(t)}^{t} \|X'(s)\|^2 ds\right)\left(\int_{s=t-j(t)}^{t} 1ds\right) + \frac{1}{2}\chi^{-1}A^2$$

$$\leq \frac{1}{2}\chi\tau\left(\int_{s=t-j(t)}^{t} \|X'(s)\|^2 ds\right) + \frac{1}{2}\chi^{-1}A^2$$

$\square$

We use this lemma twice on $\|Y\|\left\|\hat{Y} - Y\right\|$ and $\|\dot{Y}\|\left\|\hat{Y} - Y\right\|$ in equation C.7 with $\chi = 2L, A = \|Y\|$ and $\chi = 4L\eta^{-1}, A = \left\|\dot{Y}\right\|$ respectively, to yield:

$$e^{-\eta^{-1}t}E'(t) \leq -\frac{1}{4}\eta^{-1}\|Y\|^2 - \frac{1}{4}\eta\|\dot{Y}\|^2$$

$$+ L\eta^{-1}\left(L\tau D(t) + \frac{1}{4}L^{-1}\|Y\|^2\right) + L\left(2L\eta^{-1}\tau D(t) + \frac{1}{8}L^{-1}\eta\|\dot{Y}\|^2\right)$$

$$= -\frac{1}{8}\eta\|\dot{Y}\|^2 + 3L^2\eta^{-1}\tau D(t)$$

The proof of convergence is completed in Section 3.

## D  OPTIMALITY PROOF

For parameter set $\sigma, L_1, \ldots, L_n, n$, we construct a block-separable function $f$ on the space $\mathbb{R}^{bn}$ (separated into $n$ blocks of size $b$), which will imply this lower bound. Define $\kappa_i = L_i/\sigma$. We define

the matrix $A_i \in \mathbb{R}^{b \times b}$ via:

$$A_i \triangleq \begin{pmatrix} 2 & -1 & 0 & & & \\ -1 & 2 & \ddots & \ddots & & \\ 0 & \ddots & \ddots & -1 & 0 & \\ & \ddots & -1 & 2 & -1 & \\ & & 0 & -1 & \theta_i \end{pmatrix}, \text{ for } \theta_i = \frac{\kappa_i^{1/2} + 3}{\kappa_i^{1/2} + 1}.$$

Hence we define $f_i$ on $\mathbb{R}^b$ via:

$$f_i = \frac{L_i - \sigma}{4} \left( \frac{1}{2} \langle x, A_i x \rangle - \langle e_1, x \rangle \right) + \frac{\sigma}{2} \|x\|^2$$

which is clearly $\sigma$-strongly convex and $L_i$-Lipschitz on $\mathbb{R}^b$. From Lemma 8 of Lan & Zhou (2015), we know that this function has unique minimizer

$$x_{*,(i)} \triangleq \left( q_i, q_i^2, \ldots, q_i^b \right), \text{ for } q = \frac{\kappa_i^{1/2} - 1}{\kappa_i^{1/2} + 1}.$$

Finally, we define $f$ via:

$$f(x) \triangleq \sum_{i=1}^{n} f_i \big( x_{(i)} \big).$$

Now let $e(i, j)$ be the $j$th unit vector of the $i$th block of size $b$ in $\mathbb{R}^{bn}$. For $I_1, \ldots, I_n \in \mathbb{N}$, we define the subspaces

$$V_i(I) = \text{span}\{e(i, 1), \ldots, e(i, I)\},$$
$$V(I_1, \ldots, I_n) = V_1(I_1) \oplus \ldots \oplus V_n(I_n).$$

$V(I_1, \ldots, I_n)$ is the subspace with the first $I_1$ components of block 1 nonzero, the first $I_2$ components of block 2 nonzero, etc. First notice that $\text{IC}(V(I_1, \ldots, I_n)) = V(I_1, \ldots, I_n)$. Also, clearly, we have:

$$\nabla_i f(V(I_1, \ldots, I_n)) \subset V(0, \ldots, 0, \min\{I_i + 1, b\}, 0, \ldots, 0). \tag{D.1}$$

$\nabla_i f$ is supported on the $i$th block, hence why all the other indices are 0. The patten of nonzeros in $A$ means that the gradient will have at most 1 more nonzero on the $i$th block (see Nesterov (2013)).

Let the initial point $x_0$ belong to $V\big( \bar{I}_1, \ldots, \bar{I}_n \big)$. Let $I_{K,i}$ be the number of times we have had $i_k = i$ for $k = 0, \ldots, K - 1$. By induction on condition 2 of Definition 4 using equation D.1, we have:

$$x_k \in V\big( \min\{\bar{I}_1 + I_{k,1}, b\}, \ldots, \min\{\bar{I}_n + I_{k,m}, b\} \big)$$

Hence if $x_{0,(i)} \in V_i(0)$ and $k \leq b$, then

$$\big\| x_{k,(i)} - x_{*,(i)} \big\|^2 \geq \min_{x \in V_i(I_{k,i})} \big\| x - x_{*,(i)} \big\|^2 = \sum_{j=I_{k,i}+1}^{b} q_i^{2j} = \left( q_i^{2I_{k,i}+2} - q_i^{2b+2} \right) / \left( 1 - q_i^2 \right)$$

Therefore for all $i$, we have:

$$\mathbb{E}\big\| x_k - x_* \big\|^2 \geq \mathbb{E}\big\| x_{k,(i)} - x_{*,(i)} \big\|^2 \geq \mathbb{E}\left[ \left( q_i^{2I_{k,i}+2} - q_i^{2b+2} \right) / \left( 1 - q_i^2 \right) \right]$$

To evaluate this expectation, we note:

$$\mathbb{E}q_i^{2I_{k,i}} = \sum_{j=0}^{k} \binom{k}{j} p_i^j (1-p_i)^{k-j} q_i^{2j}$$

$$= (1-p_i)^k \sum_{j=0}^{k} \binom{k}{j} \left( q_i^2 p_i (1-p_i)^{-1} \right)^j$$

$$= (1-p_i)^k \left( 1 + q_i^2 p_i (1-p_i)^{-1} \right)^k$$

$$= \left( 1 - (1-q_i^2)p_i \right)^k$$

Hence

$$\mathbb{E}\|x_k - x_*\|^2 \geq \left( \left( 1 - (1-q_i^2)p_i \right)^k - q_i^{2b} \right) q_i^2 / (1-q_i^2).$$

For any $i$, we may select the starting iterate $x_0$ by defining its block $j = 1, \ldots, n$ via:

$$x_{0,(j)} = (1 - \delta_{ij})x_{*,(j)}$$

For such a choice of $x_0$, we have

$$\|x_0 - x_*\|^2 = \left\| x_{*,(i)} \right\|^2 = q_i^2 + \ldots + q_i^{2b} = q_i^2 \frac{1 - q_i^{2b}}{1 - q_i^2}$$

Hence for this choice of $x_0$:

$$\mathbb{E}\|x_k - x_*\|^2 / \|x_0 - x_*\|^2 \geq \left( \left( 1 - (1-q_i^2)p_i \right)^k - q_i^{2b} \right) / (1 - q_i^{2b})$$

Now notice:

$$\left( 1 - (1-q_i^2)p_i \right)^k = \left( q_i^{-2} - (q_i^{-2}-1)p_i \right)^k q_i^{2k} \geq q_i^{2k}$$

Hence

$$\mathbb{E}\|x_k - x_*\|^2 / \|x_0 - x_*\|^2 \geq \left( 1 - (1-q_i^2)p_i \right)^k \left( 1 - q_i^{2b-2k} \right) / (1 - q_i^{2b})$$

Now if we let $b = 2k$, then we have:

$$\mathbb{E}\|x_k - x_*\|^2 / \|x_0 - x_*\|^2 \geq \left( 1 - (1-q_i^2)p_i \right)^k \left( 1 - q_i^{2k} \right) / (1 - q_i^{4k})$$

$$= \left( 1 - (1-q_i^2)p_i \right)^k / (1 + q_i^{2k})$$

$$\mathbb{E}\|x_k - x_*\|^2 / \|x_0 - x_*\|^2 \geq \frac{1}{2} \max_i \left( 1 - (1-q_i^2)p_i \right)^k$$

Now let us take the minimum of the right-hand side over the parameters $p_i$, subject to $\sum_{i=1}^{n} p_i = 1$. The solution to this minimization is clearly:

$$p_i = (1 - q_i^2)^{-1} / \left( \sum_{j=1}^{n} (1 - q_j^2)^{-1} \right)$$

Hence

$$\mathbb{E}\|x_k - x_*\|^2/\|x_0 - x_*\|^2 \geq \frac{1}{2}\left(1 - \left(\sum_{j=1}^n (1 - q_j^2)^{-1}\right)^{-1}\right)^k$$

$$\sum_{j=1}^n (1 - q_j^2)^{-1} = \frac{1}{4}\sum_{j=1}^n \left(\kappa_i^{1/2} + 2 + \kappa_i^{-1/2}\right)$$

$$\geq \frac{1}{4}\left(\sum_{j=1}^n \kappa_i^{1/2} + 2n\right)$$

$$\mathbb{E}\|x_k - x_*\|^2/\|x_0 - x_*\|^2 \geq \frac{1}{2}\left(1 - \frac{4}{\sum_{j=1}^n \kappa_i^{1/2} + 2n}\right)^k$$

Hence the complexity $I(\epsilon)$ satisfies:

$$\epsilon \geq \frac{1}{2}\left(1 - \frac{4}{\sum_{j=1}^n \kappa_i^{1/2} + 2n}\right)^{I(\epsilon)}$$

$$I(\epsilon) \geq -\left(\ln\left(1 - \frac{4}{\sum_{j=1}^n \kappa_i^{1/2} + 2n}\right)\right)^{-1} \ln(1/2\epsilon)$$

$$= \frac{1}{4}(1 + o(1))\left(n + \sum_{j=1}^n \kappa_i^{1/2}\right)\ln(1/2\epsilon)$$

## E  EXTENSIONS

As mentioned, a stronger result than Theorem 1 is possible. In the case when $L_i = L$ for all $i$, we can consider a slight modification of the coefficients:

$$\alpha \triangleq \left(1 + (1 + \psi)\sigma^{-1/2}S\right)^{-1} \quad \text{(E.1)}$$

$$\beta \triangleq 1 - (1 + \psi)^{-1}\sigma^{1/2}S^{-1} \quad \text{(E.2)}$$

$$h \triangleq \left(1 + \frac{1}{2}\sigma^{1/2}L^{-1/2}\psi\right)^{-1}. \quad \text{(E.3)}$$

for the asynchronicity parameter:

$$y_k = \alpha v_k + (1 - \alpha)x_k, \quad \text{(E.4)}$$

$$x_{k+1} = y_k - hL^{-1}\nabla_{i_k}f(\hat{y}_k), \quad \text{(E.5)}$$

$$v_{k+1} = \beta v_k + (1 - \beta)y_k - \sigma^{-1/2}L^{-1/2}\nabla_{i_k}f(\hat{y}_k). \quad \text{(E.6)}$$

$$\psi = 6\kappa^{1/2}n^{-1} \times \tau \quad \text{(E.7)}$$

This leads to complexity:

$$K(\epsilon) = (1 + \psi)n\kappa^{1/2}\ln(1/\epsilon) \quad \text{(E.8)}$$

Here there is no restriction on $\psi$ as in Theorem 1, and hence there is no restriction on $\tau$. Assuming $\psi \leq 1$ gives optimal complexity to within a constant factor. Notice then that the resulting condition of $\tau$

$$\tau \leq \frac{1}{6}n\kappa^{-1/2} \quad \text{(E.9)}$$

now essentially matches the one in Theorem 3 in Section 3. While this result is stronger, it increases the complexity of the proof substantially. So in the interests of space and simplicity, we do not prove this stronger result.

