# OpenReview forum: "A2BCD: Asynchronous Acceleration with Optimal Complexity"
_ICLR.cc/2019/Conference_

### Official Review · AnonReviewer2 · 2018-10-28
**An elegant solution to a long-standing open question as to the speed-up of asynchronous distributed coordinate descent**

**Rating:** 9
**Confidence:** 5

**Review:**

In distributed optimisation, it is well known that asynchronous methods outperform synchronous methods in many cases. However, the questions as to whether (and when) asynchronous methods can be shown to have any speed-up, as the number of nodes increases, has been open. The paper under review answers the question in the affirmative and does so very elegantly.

I have only a few minor quibbles and a question. There are some recent papers that could be cited:
http://proceedings.mlr.press/v80/zhou18b.html
http://proceedings.mlr.press/v80/lian18a.html
https://nips.cc/Conferences/2018/Schedule?showEvent=11368
and the formatting of the bibliography needs to be improved.

In the synchronous case, some of the analyses extend to partially separable functions, e.g.:
https://arxiv.org/abs/1406.0238
and citations thereof. Would it be possible to extend the present work in that direction?

---

> ### Author Response · Authors · 2018-11-12
> **Thank you & ESO Assumption**
>
> We thank the reviewer for their kind words, and for drawing our attention to some important papers on asynchronous algorithms. We will add a discussion of these references later today. Apart from the odd spacing of the items at the end of the bibliography, are there any other formatting issues you were referring to?
>
> “Would it be possible to extend the present work in that direction?” Yes, we believe it is extremely likely that it is possible to extend our work in this direction. From a technical viewpoint: The ESO assumption leads to very similar inequalities in the partially separable case as in our base case. The only difference is that the inequalities will rely on ESO parameters instead of coordinate Lipschitz parameters. Given that, it should be possible to map our proof with proper modifications onto this setting. Moreover, we believe the ESO assumption may allow for a tighter analysis, leading to larger allowable delays. ESO has already been used to obtain tighter bounds in the mini-batch setting, which is intimately related to the asynchronous setting. Exploring the interaction between ESO and asynchronicity will surely yield interesting results.

---

### Official Review · AnonReviewer3 · 2018-10-29
**questions about theories**

**Rating:** 7
**Confidence:** 5

**Review:**

This paper studies the combination of the asynchronous parallelization and the accelerated stochastic coordinate descent method. The proved convergence rate is claimed to be consistent with the non parallel counterpart. The linear speedup is achievable when the maximal staleness is bounded by n^{1/2} roughly, that sounds very interesting result to me. However, I have a few questions about the correctness of the results:

- Theorem 1 essentially shows that every single step is guaranteed to improve the last step in the expectation sense. However, this violates my my experiences to study Nesterov's accelerated methods. To my knowledge, Nesterov's accelerated methods generally do not guarantee improvement over each single step, because accelerate methods essentially constructs a sequence z_{t+1} = A z_t where A is a nonsymmetric matrix with spectral norm greater than 1.

- The actual implemented algorithm is using the sparse update other than the analyzed version, since the analyzed version is not efficient or suitable for parallelization. However, the sparse updating rule is equivalent to the original version only for the non asynchronous version. Therefore, the analysis does not apply the actual implementation.

minors:
- pp2 line 8, K(epsilon) is not defined
- Eq. (1.4), the index is missing.
- missing reference: An Asynchronous Parallel Stochastic Coordinate Descent Algorithm, ICML 2014.

---

> ### Author Response · Authors · 2018-11-12
> **Thanks you. Monotonicity. Implementation.**
>
> We thank the reviewer for their time and their kind words. The “minors” will be addressed later today in an edit.
>
> “Theorem 1 essentially shows...” You are correct in stating that the objective function value and the distance to the solution are not guaranteed to improve in expectation at each iteration. However, in “Efficiency of coordinate descent methods on huge-scale optimization problems” Nesterov (2012), Nesterov discovered that a certain linear combination of both is guaranteed to decrease linearly and monotonically in expectation at every step (see Theorem 6 of that paper). The most familiar way to prove convergence is with estimating sequence techniques. However, we found Nesterov’s Lyapunov function approach to be a better starting point in light of the existing Lyapunov function techniques for asynchronous algorithms.
>
> As mentioned in Remark 1, our proof, Lyapunov function, and results essentially reduce to Nesterov’s proof, Lyapunov function, and results in the synchronous case where $\tau=0$. So, a guaranteed improvement of the Lyapunov function at every step should not be that surprising.
>
> “The actual implemented algorithm...” This is correct. Depending on the problem at hand and the computational architecture, a different implementation may be the most efficient. These different implementations may have slightly different convergence proofs & properties. It is unclear if there is a general way to prove convergence results for all possible implementations. So, we were forced to simply chose a base case/setup that was as similar as possible to other literature on asynchronous optimization algorithms (i.e., similar to Liu & Wright). We chose ridge regression for our experiments, even though it doesn’t exactly fit into our base case, because it was of general interest. It also demonstrates that even though this is a coordinate method, it can be used on finite-sum problems via duality. Our proof of convergence for this base case can be seen as a roadmap to prove convergence for other asynchronous accelerated algorithm implementations, which we expect to be fairly similar.
>
> We also believe that our sparse implementation in itself is a useful contribution to the field. We observed that the linear transformation of Lee & Sidford (2013) leads to “coordinate friendliness”, and hence efficient updates. This realization was essential to obtaining a state-of-the-art coordinate descent algorithm, and led to a massive speedup for both the synchronous and asynchronous case.

---

### Official Review · AnonReviewer1 · 2018-11-02
**Review: A2BCD**

**Rating:** 7
**Confidence:** 4

**Review:**

The authors design an accelerated, asynchronous block coordinate descent algorithm, which, for sufficiently small delays attains the iteration complexity of the current state of the art algorithm (which is not parallel/asynchronous). The authors prove a lower bound on the iteration complexity in order to show that their algorithm is near optimal. They also analyze an ODE which is the continuous time limit of A2BCD, which they use to motivate their approach.

I am a little bit confused about the guarantee of the algorithm, as it does not agree with my intuition. Perhaps I am simply mistaken in my intuition, but I am concerned that there may need to be additional premises to the Theorem.

My main confusion is with Theorem 1, which says that for $\psi < 3/7$ the iteration complexity is approximately the iteration complexity of NU_ACDM times a factor of $(1 + o(1))/(1-\psi)$, i.e. within that factor of the optimal *non-asynchronous/parallel* algorithm. In particular, since $\psi < 3/7$ this means that the algorithm is within a $7/4 + o(1)$ factor. As mentioned in Corollary 3, this applies for instance when $L_i = L$ for all i and $\tau = \Theta( n^{1/2}\kappa^{-1/4} )$. Therefore, in a regime where $n \approx \kappa$, and $n$ very large, this would indicate that the algorithm would be almost as good as the best synchronous algorithm even for delays $\tau \approx n^{1/4}$. Perhaps I am missing something, but this seems very surprising to me, in particular, I would expect more significant slowdown due to $\tau$.

I am also a little bit surprised that the maximum tolerable delay is proportional to the *minimum* smoothness parameter $\underbar{L}$. It seems like decreasing $\underbar{L}$ should make optimization easier and therefore more delay should be tolerated. Perhaps this is simply an artifact of the analysis.

---

> ### Author Response · Authors · 2018-11-13
> **Thank you. Intuition.**
>
> We thank the reviewer for their time and consideration of our work. We hope that we can convince the reviewer to increase their score. We believe that our work solves a difficult outstanding problem in the field of asynchronous optimization, and provides a valuable contribution to ICLR.
>
> “My main confusion…” We admit this may sound somewhat surprising. However, this result is consistent with many previous theoretical and experimental results in the coordinate descent setting. It is also consistent with the ODE analysis in our paper. For example, in “An asynchronous parallel stochastic coordinate descent algorithm” Liu & Wright (2015), authors obtain linear convergence results for asynchronous RBCD. They obtain a complexity that is within a factor of 4 of the sharp iteration complexity for the synchronous case. Experimentally though, they observe that the complexity for the asynchronous RBCD is essentially the same as serial RBCD -- even for up to 40 cores. So essentially, they observed that this penalty factor is approximately 1 -- not 4. In “More Iterations per Second, Same Quality”, Hannah & Yin manage to prove that this factor is ~1. Their condition on the maximum delay is $\tau = o(m^(½))$.
>
> Our work extends these theoretical and experimental results to the accelerated case. As you observed, for $\kappa\approx n $, our condition is slightly more restrictive at $\tau = o(m^(¼))$. In our experiments, we observed that the error vs. number of iterations for A2BCD was essentially the same for synchronous accelerated RBCD. However, we did not include graphs in the interest of space (though we can add these to the appendix if you think they would add to the paper).
>
> Let us offer some insight on why this is possible. Consider the non-accelerated case. Coordinate descent methods only modify a block of the solution vector. If one does full-gradient updates, it is known that it is impossible to obtain a speedup. However, since we are only modifying a single block at a time, it is plausible that the delayed gradient would be a good surrogate for the true gradient -- at least on average. Most of the solution vector is actually up to date, but a fraction $O(1/\sqrt{n})$ is outdated. The outdated fraction has a uniformly random distribution, which prevents blocks that have a large influence on the value of the gradient from being outdated most of the time. Given this, it makes sense that some delay is tolerable from a complexity standpoint, and that it is only a question of how much.
>
> In the accelerated case, without modification, we are applying dense updates to the solution vectors. This is because of the averaging steps. However, notice that the quantities that we are averaging are up-to-date, since they are centrally maintained (this turns out to be essential experimentally). It is only the gradient part that can be outdated. Hence, it still remains plausible that our delayed updates are a good surrogate for non-delayed updates.

---

> > ### Comment · AnonReviewer1 · 2018-11-13
> > **Thanks for the explanation**
> >
> > Having read your explanation, I am convinced that the issue was with my intuition. I see how $\underbar{L}$ changes the algorithm's step sizes, it might be good to include a mention of how you can be slightly less aggressive at the cost of a factor of 2, but perhaps that is obvious to most readers.
> >
> > Thanks for the comments, I have revised my score for the paper.

---

> ### Author Response · Authors · 2018-11-13
> **L Lower Bar**
>
> “I am also a little bit surprised...” Indeed we were also surprised :-). The reviewer brings up a very interesting point of discussion. You are correct that all things being equal, having a lower $\underbar{L}$ would make optimization easier. However, changing $\underbar{L}$ actually changes the base synchronous algorithm, and hence you can’t make a direct comparison.
>
> Though we are not completely sure, we actually believe that this is not an artifact. See (***) ahead for why we are not completely sure. A lower $\underbar{L}$ means that there are smaller coordinate Lipschitz constants. The step sizes for these small-Lipschitz coordinates are therefore more aggressive. More aggressive steps increase the error due to asynchronicity. The increase in these errors leads to a more restrictive condition on the delay. The effect of having a lower $\underbar{L}$ on the asynchronous error can be seen in the middle of page 18. This effects coordinates with larger Lipschitz constants less, because they are already taking more conservative step sizes.
>
> However, it is possible to temper this aggression to obtain a weaker condition on the maximum delay, at the cost of a constant factor in terms of the complexity. We can do this by overestimating the Lipschitz constants of the smaller-value coordinates. We did not include a discussion of this because of space limitations, and our choice to focus on the scenarios where the complexities for asynchronous and synchronous are nearly matched. Consider the $L_{½}$ average of the Lipschitz constants $L_{½} = ((1/n)sum_{i=1}^nL_i^{½})^2$. We replace $L_i$ with $max(L_i,L_{½})$. This will at most double $S$, which means the complexity may double. However because we are taking less aggressive steps on small-Lipschitz coordinates, we obtain a weaker condition on the maximum delay. Overestimating the Lipschitz constant in this way amounts to replacing $\underbar{L}$ with $L_{½}$ in the convergence condition (and replacing $S$ with $2S$). This also leads to a less counterintuitive condition on the delay.
>
> (***) It might be possible to tighten our analysis, and weaken the condition on the delay further, and perhaps change the dependence on $\underbar{L}$. This proved difficult because coordinate-smoothness is actually not that well understood. For instance, the sharp complexity of serial (non-accelerated!) coordinate descent for coordinate smooth functions is actually unknown (to our surprise) because of the absence of nontrivial lower performance bounds for RBCD. For the case where we do not assume coordinate smoothness, simply smoothness, there exist sharp worst-case analyses of coordinate descent. These were important roadmaps that we used to analyze convergence. However, no such roadmap exists in the coordinate-smooth case. There has been some interesting work in this direction by Richtarik and Takac using the so-called “expected separable overapproximation” assumption. However, the meaning and implications of this assumption are unclear. A better understanding of coordinate smoothness in the serial case may enable us to improve our analysis.
>
> We can discuss the difficulty and open questions in this setting if the reviewer is interested. We hope that the reviewer will reconsider their score of our paper.

---

### Author Response · Authors · 2018-11-13
**Revision**

Dear reviewers. We have revised our paper to include relevant references that we overlooked and minor changes that were suggested. I apologize for the lateness. I (main author) have had the flu and fever for almost a week, which makes this more challenging.

We hope that we have convinced Reviewer 1 that our results are plausible. It has been generally observed experimentally and theoretically, that at least for smooth convex optimization problems, asynchronicity does not cause significant slowdown (for delays not too large). Our results extend this to the case of an accelerated algorithm. We hope that we have helped to reconile Reviewers 3's intuition on acceleration with our results. Our monotonicity is an extension of similar results by Nesterov. We thank Reviewer 2 for drawing our attention to the rich body of work on asynchronous SGD.

In light of our responses to the reviewers' concerns, we hope that you will reconsider our current scores. We believe our work has solved a longstanding and challenging problem in asynchronous optimization. We invite the reviewers' comments on anything else that would help improve our paper and its impact on the ML community. It would be a great honor to present our work at ICML 2019. Sincerely,

The Authors

---

### Meta-Review · Area_Chair1 · 2018-12-15
**a clear accept**

**Confidence:** 4
**Recommendation:** Accept (Poster)

**Metareview:**

The reviewers all agreed that this paper makes a strong contribution to ICLR by providing the first asynchronous analysis of a Nesterov-accelerated coordinate descent method.